



# Carbon dioxide cover: carbon dioxide column concentration seamlessly distributed globally during 2009–2020

Haowei Zhang[1]★, Boming Liu[2]★, Xin Ma[2], Ge Han[3], Qinglin Yang[3], Yichi Zhang[3], Tianqi Shi[2], Jianye Yuan[1], Wanqi Zhong[2], Yanran Peng[1], Jingjing Xu[1], Wei Gong[1]

[1]School of Electronic Information, Wuhan University, Wuhan 473072, China
[2]State Key Laboratory of Information Engineering in Surveying, Mapping, and Remote Sensing, Wuhan University, Wuhan 430079, China
[3]School of Remote Sensing and Information Engineering, Wuhan University, Wuhan 430079, China
★These authors contributed to the work equally and should be regarded as co-first authors.

*Correspondence to*: XinMa (maxinwhu@whu.edu.cn)

**Abstract.** For carbon dioxide concentration ($XCO_2$) distribution, the improvement of spatial and temporal resolution is very important in some scientific studies (e.g., studies of the carbon cycle and assessment of carbon emissions based on top-down theory). However, carbon sniffing satellites based on passive theory (e.g., Gosat-2, OCO-2, and OCO-3) are susceptible to cloud and aerosol interference when the data are captured. Therefore, the data collected by carbon sniffing satellites have relatively low utilization, especially in some regions where data gaps exist. Here, we present the Carbon Dioxide Coverage (CDC) dataset, an innovative theory to obtain high spatial and temporal resolution maps of $XCO_2$ distribution by combining spatial attributes and extracted temporal attributes from the GOSAT satellite series data. This theory is divided into the following three parts. Firstly, several background values in the raw GOSAT data were removed through data pre-processing, and for spatial attributes, GOSAT satellite data gap areas were filled by combining adjacent GOSAT data and empirical Bayesian kriging (EBK) theory in the study area. Secondly, for the temporal attributes, we constructed a time profile parameter library, based on the GOSAT data of the time series to extract the temporal parameters from a specific formula at each point of the study area. Finally, for the integration of temporal and spatial information, based on the GOSAT satellite data and the populated data based on spatial attributes, we assign the temporal parameter information from the time parameter library to each pixel location in the study area, combining the transfer component analysis (TCA) theory, and then combine the assigned parameters with specific formulas to complete the prediction of $XCO_2$ distribution. For temporal resolution, both the GOSAT_FTS_L3_V2.95 and CDC datasets are monthly-averaged resolution datasets from 2010 to 2020. And for spatial resolution, the CDC dataset is 0.25° resolution with a significant improvement compared to GOSAT_FTS_L3_V2.95 which is 2.5° resolution. And the dataset contained 136 files. Besides, for the data validation part, we used OCO-2 satellite data from 2009 to 2020 and TCCON data at mid and low latitudes, respectively. This CDC dataset and the original data from the TCCON sites were compared on a monthly-averaged scale. And the results showed that $R^2$ was 0.9686, and RMSE was 1.3811 ppm. We also derived statistical monthly averaged $XCO_2$ from OCO-2 data and compared it with the data set from our theory. And our evaluation index R was greater than 0.7, by comparison with OCO-2 during 2014-2020. Finally, to assess the accuracy of





the algorithm, we compared the predicted results with the input data for the period of 2009-2020. And the comparison results show that the mean value of $R^2$ is 0.93 and the mean value of RMSE is 0.53 ppm during 2010-2020. Data gaps produced by

sniffer satellites are disturbed by factors such as clouds and aerosols and can be filled by this mapping technique is mentioned in this paper. This technique improves the utilization of $XCO_2$ and the accuracy and resolution of the CDC dataset is sufficient for scientific applications. And the CDC dataset is publicly available at https://doi.org/10.6084/m9.figshare.17826404.v4 (Zhang et al., 2022), which is of significance for a multitude of scientific carbon research.

**1 Introduction**

Global climate change exerts increased risks and impacts on natural and human life, such as rising sea levels, heat waves, floods and droughts, erosion of food security, and slowing economic growth (Field et al., 2014; Diaz et al., 2017). The skyrocketing level of greenhouse gas in recent decades is the main cause of global climate change (Black et al., 2011). Therefore, monitoring the changes in spatiotemporal carbon dioxide concentration ($XCO_2$) in the global atmosphere is crucial. The sniffer satellites in orbit at present are carrying passive detectors (Basilio et al., 2014; Nakajima et al., 2017), and the

quality of the data collected is limited by several factors, such as cloud cover, lack of observations in high-latitude areas and at night, and sensitivity to aerosols. Therefore, the acquisition of spatio-temporal maps of $XCO_2$ distribution with high accuracy and resolution is essential to facilitate the study of the carbon cycle, carbon sources, carbon sinks, carbon neutrality, and carbon emissions assessed through top-down theory.

Scientists have conducted downscaling studies on carbon detection series satellite data. Tomasada et al. (2009;2008) and Liu

et al. (2012) generated monthly-averaged $CO_2$ distribution maps by using ordinary kriging interpolation of GOSAT Level 2 (L2) products. Hammerling et al. (2012) obtained $CO_2$ maps mainly by processing simulated satellite observations by using a moving kriging window. Mueller et al. (2008) reconstructed global monthly-averaged $CO_2$ fluxes from ground observations by using the geostatistical inverse modeling theory. Moreover, Katzfuss et al. (2011;2012) completed spatiotemporal smoothing of global $XCO_2$ data, the theory of which focused on a fully Bayesian hierarchical approach. Zeng et al. (2013)

proposed a spatiotemporal kriging theory, applied it to model GOSAT data in China, and obtained the monthly-averaged distribution of $XCO_2$.

The interpolation method commonly used for satellite $XCO_2$ observations is the conventional geostatistical spatial prediction method, which considers spatial autocorrelation only (Tomosada et al. 2009; Tomosada et al. 2008; Liu et al. 2012). This method requires a long time series of data so as to ensure sufficient data for stable variometric estimations, but it ignores the

time structure in the data. In addition, on the basis of spatial interpolation, several scholars further integrated time information into the interpolation method and obtained good results (Zeng et al. 2013; Yang et al. 2020; Gribov et al. 2012; Ma et al. 2021). Although these methods produce good results from a mathematical point of view, in studies that utilized these methods, the prior time profile information of $XCO_2$ was rarely considered, resulting in insufficient adjustment of the temporal information, as reflected by the large differences between the monthly-averaged $XCO_2$ and the true value. Therefore, in this study, we





integrate the prior information of the original data into a new spatiotemporal interpolation theory that considers the time variation of concentration distribution to effectively improve data accuracy. In other words, we propose a new method to improve the utilization of $XCO_2$ data. First, several background values in the raw GOSAT data were removed through data pre-processing, and for spatial attributes, GOSAT satellite data gap areas were filled by combining adjacent GOSAT data and empirical Bayesian kriging (EBK) theory in the study area. Secondly, for the temporal attributes, we constructed a time profile

parameter library, based on the GOSAT data of the time series to extract the temporal parameters from a specific formula at each point of the study area. Finally, for the integration of temporal and spatial information, based on the GOSAT satellite data and the populated data based on spatial attributes, we assign the temporal parameter information from the time parameter library to each pixel location in the study area, combining the transfer component analysis (TCA) theory, and then combine the assigned parameters with specific formulas to complete the prediction of $XCO_2$ distribution.

The focus of this work is to provide a global dataset of the monthly-averaged $XCO_2$ at 0.25° based on the theory presented in the paper and the discrete $XCO_2$ measured by the GOSAT satellite. The CDC dataset extends from 2009 to 2020 and from 50° S to 50° N.  The validation of the CDC dataset will be performed by comparing it with those from OCO-2, TCCON, and the input GOSAT dataset (which was not involved in the generation of the CDC dataset). Namely, the accuracy validation of the CDC dataset is divided into the following parts in this paper. First, based on the theory proposed in this work and the

GOSAT_L3 data, we compare the spatio-temporal prediction data generated in each TCCON site with the data from the corresponding TCCON site. Second, we derived statistical monthly-averaged $XCO_2$ from OCO-2 data and compared it with the data set from our theory. Finally, to assess the accuracy of the algorithm, we compared the results of the model predictions with the input data for the period 2009-2020.

The advantages of the global CDC dataset are (1) its large spatial coverage (From approximately 55° S to 55° N with a

resolution of 0.25°) and (2) 12-year time series (Monthly-averaged $XCO_2$ from 2009 to 2020). Thus, the CDC dataset can be used to study the global $XCO_2$ at timescales ranging from seasons to decades and from cities to countries.  Besides, the $XCO_2$ data calculated by the model presented in this paper can be input into the atmospheric chemical transport model and can also contribute to the study of the carbon cycle. And the satellite data of global observations (such as OCO-2, OCO-3, GOSAT, GOSAT-2 and Tansat) have been widely used for the calculation of global carbon sources and sinks. Therefore, this technique

improves the utilization of $XCO_2$ and the accuracy and resolution of the CDC dataset is sufficient for scientific applications. Considering that GOSAT can help to obtain $XCO_2$ at the globle scale, whose data is used as the primary dataset in this work. It enables the development of strategies to reduce $XCO_2$ at the global scale. The dataset and related codes are publicly available at https://doi.org/10.6084/m9.figshare.17826404.v4 (Zhang et al., 2022), which are of significance for a multitude of scientific research and applications.



## 2 Materials and Methods

### 2.1 Data description

The time span of GOSAT satellite data (2009–2020) is longer than that of OCO-2, OCO-3, and Tansat. Thus, we selected the bias-corrected data of GOSAT_FTS_L3_V2.95. And the accuracy of the comparison between the GOSAT data product and the TCCON site was 0.56 ppm (Noël et al. 2021; Watanabe et al. 2015;). And the GOSAT orbits at an altitude of approximately 666 km, with 10.5 km of spatial resolution and three-day temporal resolution. The time resolution of GOSAT-2 satellite is 6 days, IFOV is 9.7km.

The GOSAT and GOSAT-2 satellites have been operational since 2009 and 2018, respectively, and the data collected by the GOSAT-1/2 satellites have the potential to reveal new information on the carbon cycle. Studies of the carbon cycle have been carried out based on atmospheric chemistry models. Such models usually require input of measured $XCO_2$ data to constrain the atmospheric chemistry model. However, the OCO-2_L2_Lite_FP9r provides data locations that are gradually shifted over time by satellite observations. And the GOSAT_L3 product only provides a long time series of cumulative observations for a fixed location, thus large vacant data areas exist in the global for the GOSAT_L3 product. Our proposed monthly-averaged $XCO_2$ map can complete the carbon cycle input on a large scale spatially and over a long time series. Therefore, our monthly-averaged $XCO_2$ map is helpful for carbon cycle studies. Because the six data channels of the sensor carried by the GOSAT satellite operate in the near-infrared part of the solar spectrum, the GOSAT satellite cannot collect data when the Earth reflects little sunlight, such as in polar regions during winter. For additional instrument's information, readers may refer to http://www.gosat.nies.go.jp/en/about_2_observe.html. Furthermore, the GOSAT-1/2 satellite provides column-averaged $XCO_2$ by measuring the spectrum reflected by sunlight in the infrared region over a global scale. However, the interference of clouds and aerosols offen results in a sparse spatiotemporal coverage for $XCO_2$ products of GOSAT.

### 2.2 Validation data

To evaluate the accuracy of the monthly-averaged $XCO_2$ data from our algorithm, we used global data of the Total Carbon Column Observing Network (TCCON) during 2009-2020. TCCON ( Iraci et al. 2017; Dubey et al. 2017; Wennberg et al. 2017; Dubey et al. 2017; Blumenstock et al. 2017; Feist et al. 2017; Warneke et al. 2017; Sussmann et al. 2017; Sussmann et al. 2017; Petri et al. 2017; Maziere et al. 2017; Morino et al. 2017; Goo et al. 2017; Shiomi et al. 2017; Morino et al. 2017; Morino et al. 2017; Griffith et al. 2017; Pollard et al. 2017; Sherlock et al. 2017; Liu et al. 2017; Wennberg et al. 2017; Wennberg et al. 2017;) is composed of ground-based Fourier transform spectrometers that record direct solar spectra in the near-infrared spectral region. And we show the global distribution of TCCON sites in Figure 1. The spectrometer used in TCCON can provide accurate and precise column-averaged abundances of $CO_2$. And the results showed that $R^2$ was 0.9686, and RMSE was 1.3811. We also collected the original $XCO_2$ from OCO-2 for comparison and to obtain abundant observations of $XCO_2$ from OCO-2 in a large range. And our evaluation index R was greater than 0.7, by comparison with OCO-2 during 2014-2020.





### 2.3 Theoretical framework

The framework in Figure 2 depicts the general methodology. The method is divided into three parts: Spatial Prediction Through EBK Theory, Prior Time Curve Parameter Library, and Integration of Temporal Attributes through TCA Theory, respectively.

In the Spatial Prediction Through EBK Theory section, the $XCO_2$ gaps are filled in the spatial attributes through EBK Theory. In the Prior Time Curve Parameter Library section, a time profile parameter library is constructed to express the temporal attributes. In the Integration of Temporal Attributes through TCA Theory section, temporal and spatial information is integrated based on TCA theory, and then combine the assigned parameters with specific formulas to complete the prediction of $XCO_2$ distribution.

### 2.3.1 Spatial Prediction Through EBK Theory

To obtain the distributions of monthly-averaged $XCO_2$ in the study area, we performed monthly-averaged calculations on the raw GOSAT_L3 data on the basis of a 0.25° grid for each month separately. Then, we used the existing EBK method to fill the areas not covered effectively and reasonably by the GOSAT data in each month. EBK can automatically perform the most difficult steps in the process of building an effective kriging model (Gribov et al. 2012; Krivoruchko et al.). The EBK can

automatically calculate parameters through the process of constructing subsets and simulations, while the same type of Kriging interpolation requires manual adjustment of parameters to receive accurate results (Krivoruchko et al. 2012; Krivoruchko et al.2019). And weighted least squares is used to estimate the semi-variogram in the same type kriging interpolation method, but the parameters in EBK are estimated by the limited maximum likelihood method. And the EBK method differs from other kriging methods in that other kriging methods assume that the estimated semi-variograms is the true semi-variograms of the

interpolated region, and use single variogram to predict the value of the unknown location. But the EBK method estimates the error of the semi-variograms. And the EBK theory will be more accurate compared to other kriging theories because it takes into account the uncertainty involved in the estimation of the semi-variogram (Pilz et al.2007). Therefore, we choose EBK interpolation method as the data processing method.

### 2.3.2 Prior Time Curve Parameter Library

To fill the region of data gaps in space, we use EBK theory in Section 2.3.1. However, this EBK theory only considers the adjacent $XCO_2$ data in the current month at the data gap location. Then, using only a theory based on spatial attributes to fill the data gap locations would have a problem: the relationship between $XCO_2$ data at adjacent times is cut off from a continuous time scale. Thus, based on EBK theory, data gap filling may result in the current month being anomalous relative to $XCO_2$ at adjacent times.

For this reason, we constructed a time profile parameter library, based on the GOSAT data of the time series to extract the temporal parameters from a specific formula at each point of the study area. We used GOSAT_L3 data as the input to build the time curve library because the data of GOSAT_L3 stably provide the monthly-averaged $XCO_2$ data of successive months





at the global scale. Furthermore, we used Eq. (1) to express the time change of XCO₂ and to fit the GOSAT_L3 data for obtaining the parameters a and b.

$$F(t) = a + b*t + c*\cos\left(\frac{2\pi t}{f}\right) + d*\sin\left(\frac{2\pi t}{f}\right) + e*\cos\left(\frac{2\pi t}{f}\right) + g*\sin\left(\frac{2\pi t}{f}\right), \tag{1}$$

where $a$ refers to the yearly averaged XCO₂; $c$, $d$, $e$, and $g$ are the coefficients of the seasonal component; $b$ is the coefficient of the interannual component; $f$ is the sampling frequency ($f = 12$ for a year); and $t$ is the sampling interval.

### 2.3.3 Integration of Temporal Attributes through TCA Theory

To fill the XCO₂ gap region, we used the EBK theory based on spatial attributes in Section 2.3.1, and constructed a time profile parameter library based on temporal attributes in Section 2.3.2. But the problem is: how to merge the parameters representing temporal attributes in the time profile parameter library with the filled XCO₂ gaps based on spatial attributes (namely, the EBK theory)? The TCA theory solves the allocation problem of parameters $b$ and $c$, which represent the XCO₂ time profile of the whole research area in the time curve parameter library. First, TCA assumes the same conditional distribution of source and target domains. second, maps the data into high-dimensional reproducing kernel Hilbert space, and then uses maximum mean discrepancy to find a mapping matrix that minimizes the marginal distribution between different domains to increase the source domain the similarity with the target domain. finally, use the data of the source and target domains and the mapping matrix to train the classifier and complete the labeling of the target domain. The core of TCA is to find a mapping matrix that satisfies the conditions (Dong et al. 2021; Pan et al. 2010; Dong et al. 2020; Dong et al. 2017).

In this study, for the spatial point locations corresponding to the temporal profile parameter library, the fitted data are set as the source domain based on Eq (1). And, the spatial interpolation data are set as the target domain based on EBK theory in the study area. Thus, each temporal profile was distributed from the source domain to the corresponding target domain based on TCA theory transfer learning. Each pixel was again fitted based on Equation 1, combined with the time-adjusted parameters $c$, $d$, $e$, and $g$, assigned by TCA theory in the target domain from the temporal profile parameter library, in order to obtain the remaining parameters $a$ and $b$. And the final fitted data represent the spatio-temporal interpolation data.

### 2.4 Accuracy Assessment

The accuracy verification process was divided into three main parts. First, this CDC dataset and the original data from the TCCON sites were compared on a monthly-averaged scale. Second, we derived statistical monthly-averaged XCO₂ from OCO-2 data and compared it with the data set from our theory. Finally, to assess the accuracy of the algorithm, we compared the results of the model predictions with the input data for the period 2009-2020.

To quantify the rationality of the proposed theory in this paper, the coefficient of determination ($R^2$) and the root mean square error (RMSE) are chosen in this manuscript. The $R^2$ can be used to evaluate the linear correlation between the results and the actual values. The RMSE is used to evaluate the bias of the prediction. The RMSE and $R^2$ can be defined as follows:



$$RMSE = \sqrt{\frac{1}{N}\sum_{i=1}^{N}|P_i - R_i|^2}, \quad (2)$$

$$\overline{y} = \frac{1}{N}\sum_{i=1}^{N}P_i, \quad (3)$$

$$R^2 = 1 - \frac{\sum_{i=1}^{N}(P_i - R_i)^2}{\sum_{i=1}^{N}(P_i - \overline{y})^2}, \quad (4)$$

where $N$ is the number of prediction locations, $P_i$ is the predicted value, and $R_i$ is the observed value.

## 3 Results

### 3.1 Evaluation using TCCON observations

The monthly-average $XCO_2$ distribution of the CDC dataset products from 2010 to 2020 is presented from Figure 7 to Figure 17. And the data dictionary corresponding to the products we show in Table 3. Considering the range of $XCO_2$ predicted by the algorithm, we matched the predicted monthly-averaged $XCO_2$ from our algorithm with the measured monthly-averaged $XCO_2$ from TCCON sites at low- and mid-latitudes at the global scale from 2009 to 2020. Then, we used two mathematical indicators ($R^2$ and RMSE) to quantitatively evaluate our algorithm. Table 1 lists the statistics for the predicted $XCO_2$ and TCCON-observed $XCO_2$. Our algorithm's $R^2$ is above 0.95, and its RMSE is below 1.5 in most of the individual TCCON sites. We also comprehensively analyzed the predicted data in 24 TCCON sites. The results showed that $R^2$ was 0.9686, and RMSE was 1.3811. Pearson's correlation coefficient was adopted to evaluate the relationship between the predicted $XCO_2$ and the $XCO_2$ from TCCON sites. We annotated P<0.01 in Table 1 to indicate that the data have a strong statistical correlation. Figure 3 shows the predicted $XCO_2$ and $XCO_2$ observations at 23 TCCON sites. Compared with other similar works, the overall evaluation metric RMSE for our product data was 1.38 and improved by 22.9 % (Li et al. 2022; Zeng et al. 2014), and the spatial resolution (0.25°) became more refined compared with the mainstream spatial resolution of 1°. The time span of the data set is 12 years from 2009 to 2020. Therefore, our data set fully satisfies the calculations of carbon sources, sinks, and emissions in a long time series.

### 3.2 Evaluation using OCO-2 observations

To evaluate the accuracy of the algorithm's predicted data at the global scale, we considered another greenhouse gas satellite, OCO-2, from the United States. OCO-2 and GOSAT satellites are $XCO_2$ monitoring satellites that use the passive inversion mode. Although the sensors onboard the two satellites are different, the data from both are a measure of $XCO_2$ columns. Several scholars have compared $XCO_2$ data from OCO-2 and GOSAT-2 and concluded that the observed data values of the two satellites are consistent and smooth (Liang et al. 2017). For these reasons, we selected measured $XCO_2$ data from OCO-2 as a comparison for verification. By doing so, we can verify our products in a wider range and with more data than fixed TCCON sites. We removed bad data in accordance with the data quality label provided by OCO-2 and obtained the monthly-



averaged XCO₂ data through statistics. The statistical results showed that all R values were greater than 0.7, and a significant correlation was observed at the 0.01 level (Table 2). OCO-2 data services were opened and closed in 2014 and 2020, respectively, and we could only obtain partial OCO-2 data. Therefore, our evaluation index (R) is relatively low in 2014 and 2020 due to the insufficient data volume. Accordingly, a density scatter diagram of each year is drawn in Figure 4, and most

of the data are distributed on the 1:1 line. The color change from blue to red in Figure 4 indicates a gradual increase in data overlap. Furthermore, the comparison results distributed near the 1:1 line are high-density data in Figure 4. The comparison of OCO-2 data during 2014–2020 revealed that our data have high accuracy and stability. We found multiple parts per million deviations present between the OCO-2 and algorithm products in Figure 4, which is due to the difference in revisit period. Compared to the revisit period of 16 days for OCO-2, the repeat period of GOSAT-2 satellite is 6 days. Therefore, GOSAT

will can sample more data than OCO-2 in a month's time. Besides, the official algorithms of OCO-2 and GOSAT-2 products are different, so the model results generated based on GOSAT-2 data will produce multiple parts per million deviations compared to the OCO-2 product during 2015-2019 period.

### 3.3 Evaluation using GOSAT_L3 observations

To assess the accuracy of the algorithm, we compared the results of the model predictions with the input data for the period

2009-2020. To validate evenly globally, we removed one column of GOSAT_L3 data for each 20° longitude interval. The removed data will be used as the validation set for validation. And this $R^2$ and RMSE are used as evaluation metrics to evaluate the validation set and the predicted data. Besides, we show the validation results of the CDC dataset according to the year interval in Figure 5 from 2009 to 2020. And the comparison results show that the mean value of $R^2$ is 0.93 and the mean value of RMSE is 0.53 ppm during 2010-2020. This indicates that the accuracy of our data products is recognized from the

GOSAT_L3 input data. In Figure 6, we show the errors for each year of predicted data. And, the fluctuations of the error bands shaded in Figure 6 are small, which indicates that the errors of the data set are in a stable state from 2010 to 2020. Because of the data from June to December in 2009, the low precision metrics indicate that the model is not suitable for incomplete years. In general, products from our models can fill the vacant areas of XCO₂ globally. As described in the theory section, our method required the input of 12 consecutive months of XCO₂ data to make predictions, with the ideal data input period being from

January to December. Because satellite observations are missing in some years (e.g., 2009, 2014, and 2015), we need to combine adjacent months to complete a continuous time period of data input. Therefore, the temporal structure of this data input may have an impact on the accuracy of the model.

### 3.4 Evaluating Dataset Uncertainty

We divided the data uncertainty into three categories. This label '1' indicates that there are no satellite observations at the

location where the label is located, and that the spatial and temporal properties of this location are adjusted. In other words, the error of the data product at the position indicated by label '1' may be the largest in the three types of labels. This label '2' represents the presence of GOSAT-2_L3 observations at the location where the label is located, but with the temporal attribute

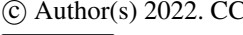


adjusted. That is, the error of the data product at the location indicated by label '2' may be small, and this kind of data has medium error in the three types of labels. This label '3'represents the presence of GOSAT-2_L3 data from satellite observations
at the location where the label is located, and the data from this location are used to build the data for the time profile library. That is, the data product error at the position indicated by label '3' is minimal. Finally, we added a data layer 'uncertainty' to show the uncertainty in the latest dataset.

## 4 Data and code availability

Version 3 of the CDC Database is available in h5 format at https://doi.org/10.6084/m9.figshare.17826404.v4 (Zhang et al.,
2022). For the data extraction approach and the data dictionary of the CDC dataset, we provide the ReadMe.pdf file in the CDC dataset repository. For the introduction of data extraction in the ReadMe.pdf file, it contains Panoly and HDFView software as well as read examples through python platform. TCCON dataset can be accessed https://tccondata.org/. Because the CDC dataset range is covered at mid to low latitudes, we match the latitude range of the CDC dataset (namely, approximately from [55N, 55S]) with the corresponding TCOON site data on the TCCON website. Besides, the OCO-2 dataset
can be accessed https://disc.gsfc.nasa.gov/datasets?page=1&keywords=OCO-2. And, the OCO-2 data version number used in the validation set is OCO2_L2_Lite_FP 9r. The compressed code has been uploaded to the repository and the file name is Code.zip in the RawDataAndCode folder of the data repository.

## 5 Conclusions

In this paper, we propose a new method to improve the utilization of $XCO_2$ data (as shown in Figure 7). First, several
background values in the raw GOSAT data were removed through data pre-processing, and for spatial attributes, GOSAT satellite data gap areas were filled by combining adjacent GOSAT data and empirical Bayesian kriging (EBK) theory in the study area. Secondly, for the temporal attributes, we constructed a time profile parameter library, based on the GOSAT data of the time series to extract the temporal parameters from a specific formula at each point of the study area. Finally, for the integration of temporal and spatial information, based on the GOSAT satellite data and the populated data based on spatial
attributes, we assign the temporal parameter information from the time parameter library to each pixel location in the study area, combining the transfer component analysis (TCA) theory, and then combine the assigned parameters with specific formulas to complete the prediction of $XCO_2$ distribution.

Besides, we evaluated the accuracy of the algorithm through three parts. First, this CDC dataset and the original data from the TCCON sites were compared on a monthly-averaged scale. And the results showed that $R^2$ was 0.9686, and RMSE was 1.3811;
Second, we derived statistical monthly-averaged $XCO_2$ from OCO-2 data and compared it with the data set from our theory. And our evaluation index R was greater than 0.7, by comparison with OCO-2 during 2014-2020; Finally, to assess the accuracy

of the algorithm, we compared the results of the model predictions with the input data for the period 2009-2020. And the comparison results show that the mean value of $R^2$ is 0.93 and the mean value of RMSE is 0.53 ppm during 2010-2020.

In general, we obtained $XCO_2$ based on GOSAT-2 data that can accurately fill the $XCO_2$ gap region in the global through the
model presented in this paper from 2009 to 2020. And for the data coverage, our data area mainly covers the middle and low latitudes in the global. Besides, the $XCO_2$ data calculated by the model presented in this paper can be input into the atmospheric chemical transport model and can also contribute to the study of the carbon cycle. And the satellite data of global observations (such as OCO-2, OCO-3, GOSAT, GOSAT-2 and Tansat) have been widely used for the calculation of global carbon sources and sinks. This mapping technique with high accuracy and resolution can fill the spatiotemporal gaps in satellite measurement,
which can meet the needs of scientific applications. And the GOSAT is the primary dataset being used in this work. It enables the development of strategies to reduce $XCO_2$ at the global scale.

## 7 Author contributions

HZ, XM and GH designed the research and developed the whole methodological framework; BL supervised the CDC dataset; QY, YZ and JY collects data for validation; HZ wrote the original draft of the manuscript; WZ, YP, JX and WG revised the
manuscript.

## 8 Competing interests

The authors declare no competing interests.

## 9 Acknowledgements

This work was supported by the National Natural Science Foundation of China (Grant No. 42171464, 41971283, 41801261,
41827801 and 41801282), the National Key Research and Development Program of China (2017YFC0212600), The Key Research and Development Project of Hubei Province(2021BCA216). The numerical calculations in this paper have been done on the supercomputing system in the Supercomputing Center of Wuhan University.

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

**Table 1.** Geographic locations of TCCON sites used for validation and the statistics used to compare predicted $XCO_2$ and TCCON $XCO_2$ observations.

| Tccon sites (Site abbreviations) | Longitude | Latitude | $R^2$ | RMSE |
|---|---|---|---|---|
| Jet Propulsion Laboratory (JC) | -118.18 | 34.20 | 0.98** | 1.07 |
| Caltech (CI) | -118.13 | 34.14 | 0.97** | 0.95 |
| Edwards (DF) | -117.88 | 34.96 | 0.98** | 0.82 |
| Four Corners (FC) | -108.48 | 36.80 | 0.96** | 0.31 |
| Lamont (OC) | -97.49 | 36.60 | 0.98** | 1.04 |
| Park Falls (PA) | -90.27 | 45.94 | 0.98** | 1.24 |
| Manaus (MA) | -60.60 | -3.21 | 0.88** | 0.64 |
| Izana (IZ) | -16.48 | 28.30 | 0.98** | 1.18 |
| Ascension Island (AE) | -14.33 | -7.92 | 0.94** | 0.93 |
| Orléans (OR) | 2.11 | 47.97 | 0.99** | 0.95 |
| Zugspitze (ZS) | 10.98 | 47.42 | 0.92** | 1.52 |
| Garmisch (GM) | 11.06 | 47.48 | 0.98** | 1.05 |
| Nicosia (NI) | 33.38 | 35.14 | 0.93** | 0.73 |
| Réunion Island (RA) | 55.49 | -20.90 | 0.96** | 1.23 |
| Hefei (HF) | 117.17 | 31.90 | 0.87** | 1.51 |
| Burgos (BU) | 120.65 | 18.53 | 0.89** | 1.01 |
| Anmeyondo (AN) | 120.65 | 36.54 | 0.90** | 1.20 |
| Saga (JS) | 130.29 | 33.24 | 0.97** | 1.26 |
| Edwards (DB) | 130.89 | -12.43 | 0.99** | 0.75 |
| Tsukuba (TK) | 140.12 | 36.05 | 0.91** | 1.89 |
| Rikubetsu (RJ) | 143.77 | 43.46 | 0.95** | 1.17 |
| Wollongong (WG) | 150.88 | -34.41 | 0.99** | 0.82 |
| Lauder01&02&03 (LL) | 169.68 | -45.04 | 0.97** | 1.44 |
| All sites | - | - | 0.97** | 1.38 |

** At the 0.01 level (two-tailed), the correlation is significant.




**Table 2**. Statistics for predicted monthly-averaged $XCO_2$ and OCO-2 monthly-averaged $XCO_2$ observations.

| Year | R | Nums |
|------|------|--------|
| 2014 | 0.37** | 129089 |
| 2015 | 0.74** | 586906 |
| 2016 | 0.75** | 789007 |
| 2017 | 0.75** | 641161 |
| 2018 | 0.70** | 768083 |
| 2019 | 0.70** | 768083 |
| 2020 | 0.72** | 28564 |

** At the 0.01 level (two-tailed), the correlation is significant.








**Table 3.** The data dictionary for the CDC dataset

| Number | field names | Data type | Unit | Further description |
|---|---|---|---|---|
| 1 | Latitude | Matrix | (Degrees, minute) | Point of latitude |
| 2 | Longitude | Matrix | (Degrees, minute) | Point of longitude |
| 3 | Spatial $XCO_2$ | Matrix | ppm | The result of spatial interpolation |
| 4 | Spatiotemporal $XCO_2$ | Matrix | ppm | The result of spatio-temporal interpolation |
| 5 | Parment a | Matrix | - | Model parameter a |
| 6 | Parment b | Matrix | - | Model parameter b |
| 7 | Parment c | Matrix | - | Model parameter c |
| 8 | Parment d | Matrix | - | Model parameter d |
| 9 | Parment e | Matrix | - | Model parameter c |
| 10 | Parment g | Matrix | - | Model parameter g |
| 11 | RMSE | Matrix | - | Model evaluation index |
| 12 | R Square | Matrix | - | Model evaluation index |
| 13 | Code Version | Float | - | Code version |
| 14 | Spatial Resolution | Float | - | Spatial resolution |
| 15 | Numbers of valid months | Int | - | Number of valid months in a year |
| 16 | Labels TCA | Int | - | Label |
| 17 | Uncertain | Int | - | Uncertain label |
| 18 | Time Curve Parameter Library | Matrix | - | Spatial position coordinates in the time curve parameter library |






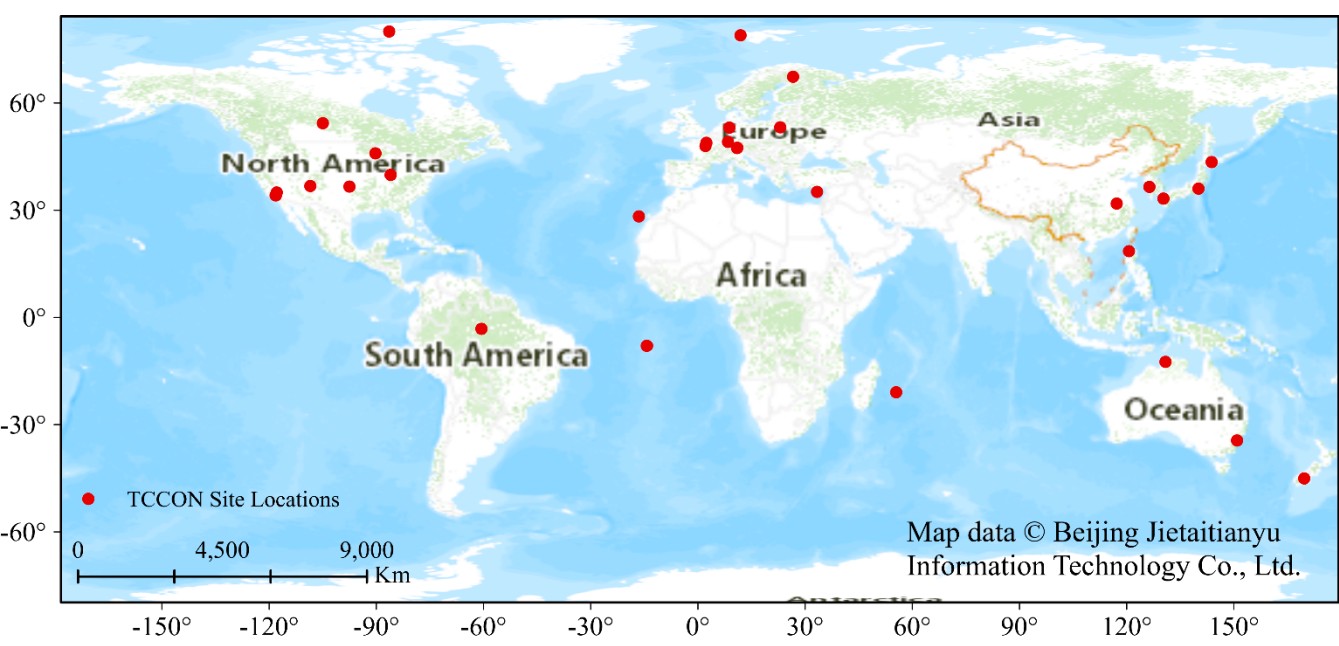

**Figure 1**. Map showing the location of TCCON sites in global.






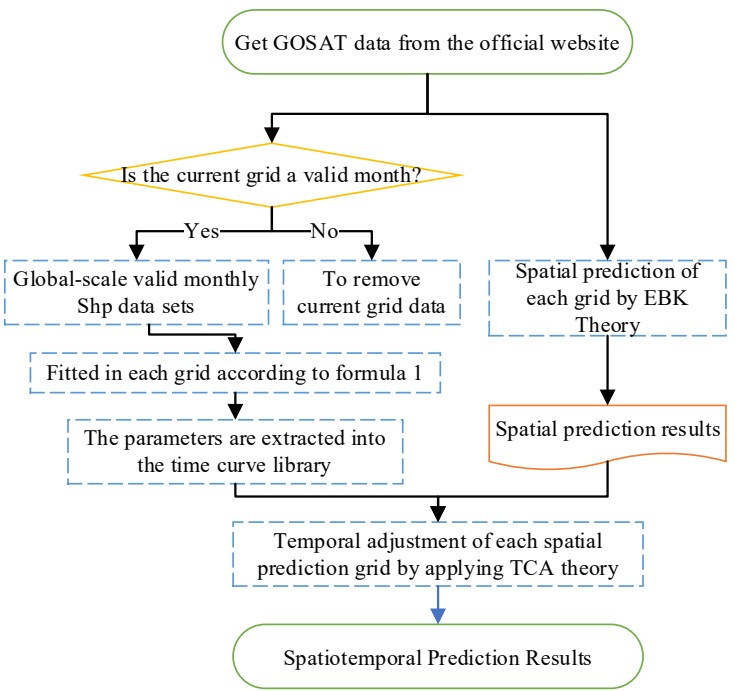

**Figure 2**. Framework of the proposed methodology.






**Figure 3.** Scatter plots of predicted XCO₂ and XCO₂ observations at 23 TCCON sites. P XCO$_2$ is the predicted XCO$_2$. T XCO$_2$ is the TCCON XCO$_2$.



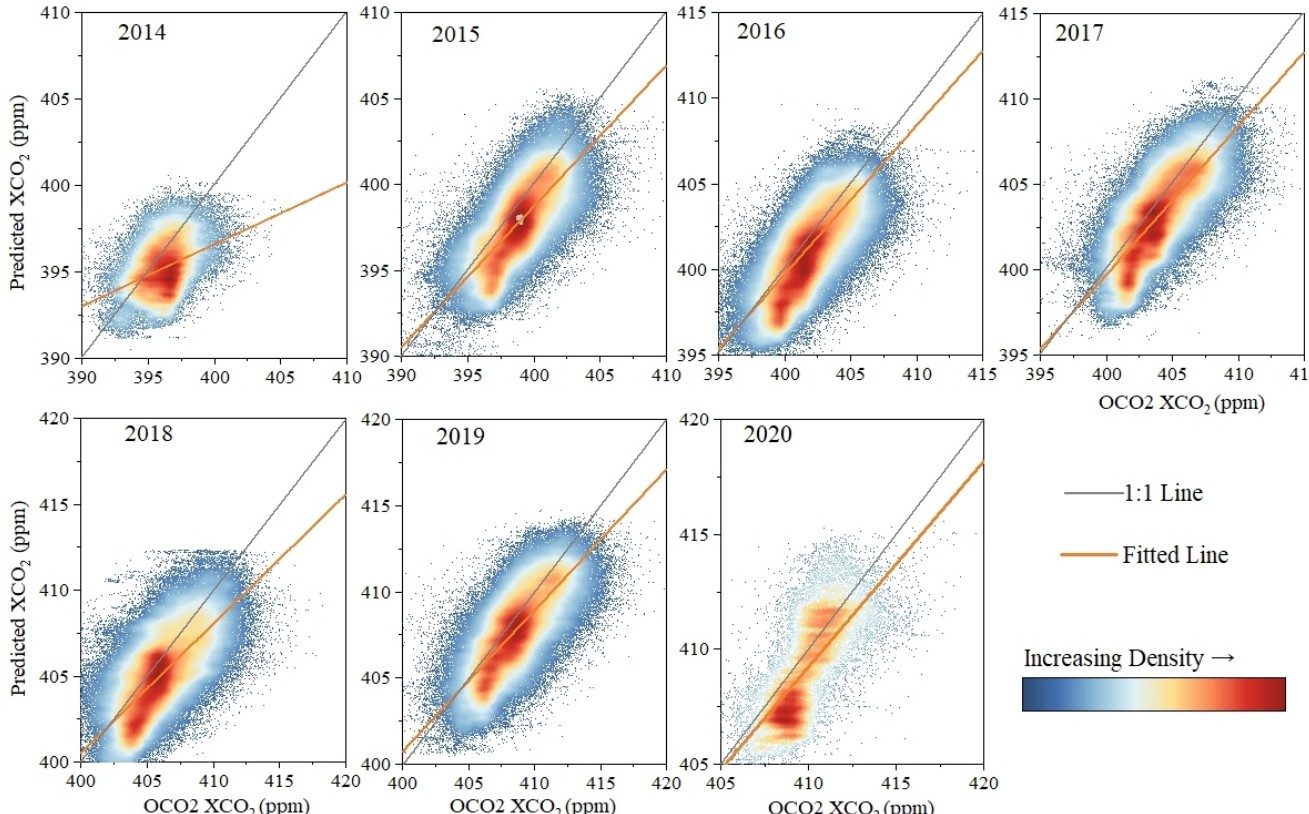

**Figure 4.** Density scatter plots of predicted XCO₂ and observed one from OCO-2.






**Figure 5.** Scatter plots of predicted XCO₂ and XCO₂ observed from GOSAT_L3. P XCO₂ is the predicted XCO₂. T XCO₂ is the GOSAT_L3 XCO₂. The blue dots in the graph represent the raw data in different years. The yellow line represents the line

where the original data was fitted.



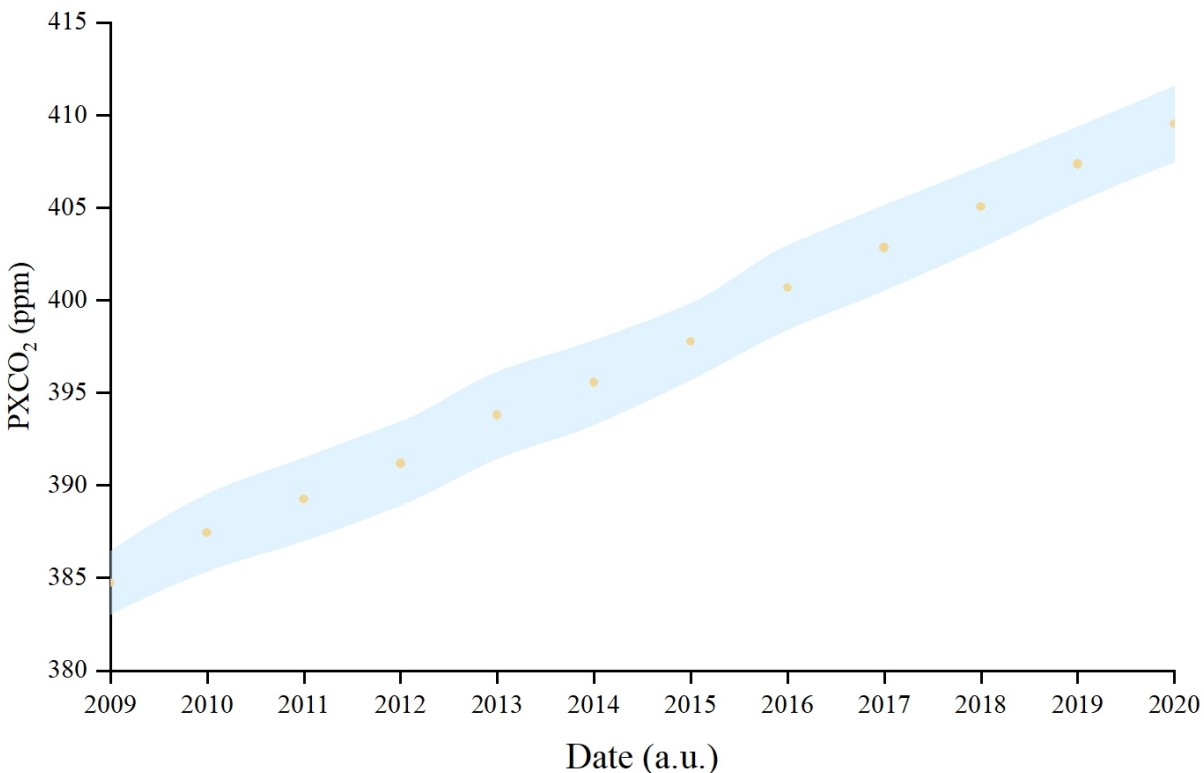

**Figure 6.** Error with graph of predicted $XCO_2$ from 2009 to 2020. $PXCO_2$ is the predicted $XCO_2$. The blue shading represents the standard deviation of the CDC data set for the screened locations in the corresponding year in the figure. The yellow dots

represent the mean of the CDC data set for the screened locations in the corresponding year in the figure.





**Figure 7.** Product data from Our Algorithm in 2010. This CDC dataset covers approximately from 55°N to 55°S, with a spatial
resolution of 0.25°.

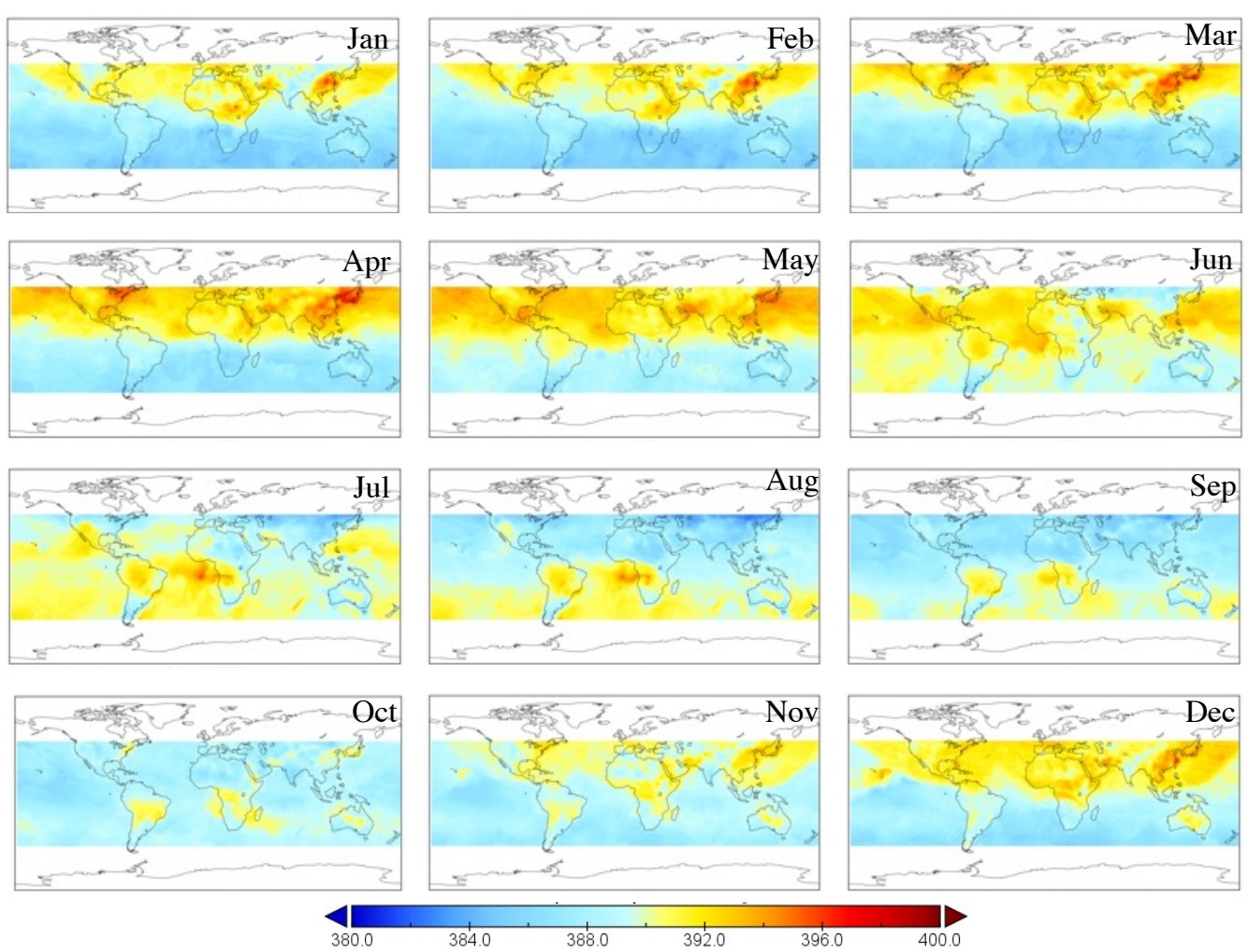


**Figure 8.** Product data from Our Algorithm in 2011. This CDC dataset covers approximately from 55°N to 55°S, with a spatial resolution of 0.25°.





**Figure 9.** Product data from Our Algorithm in 2012. This CDC dataset covers approximately from 55°N to 55°S, with a spatial resolution of 0.25°.



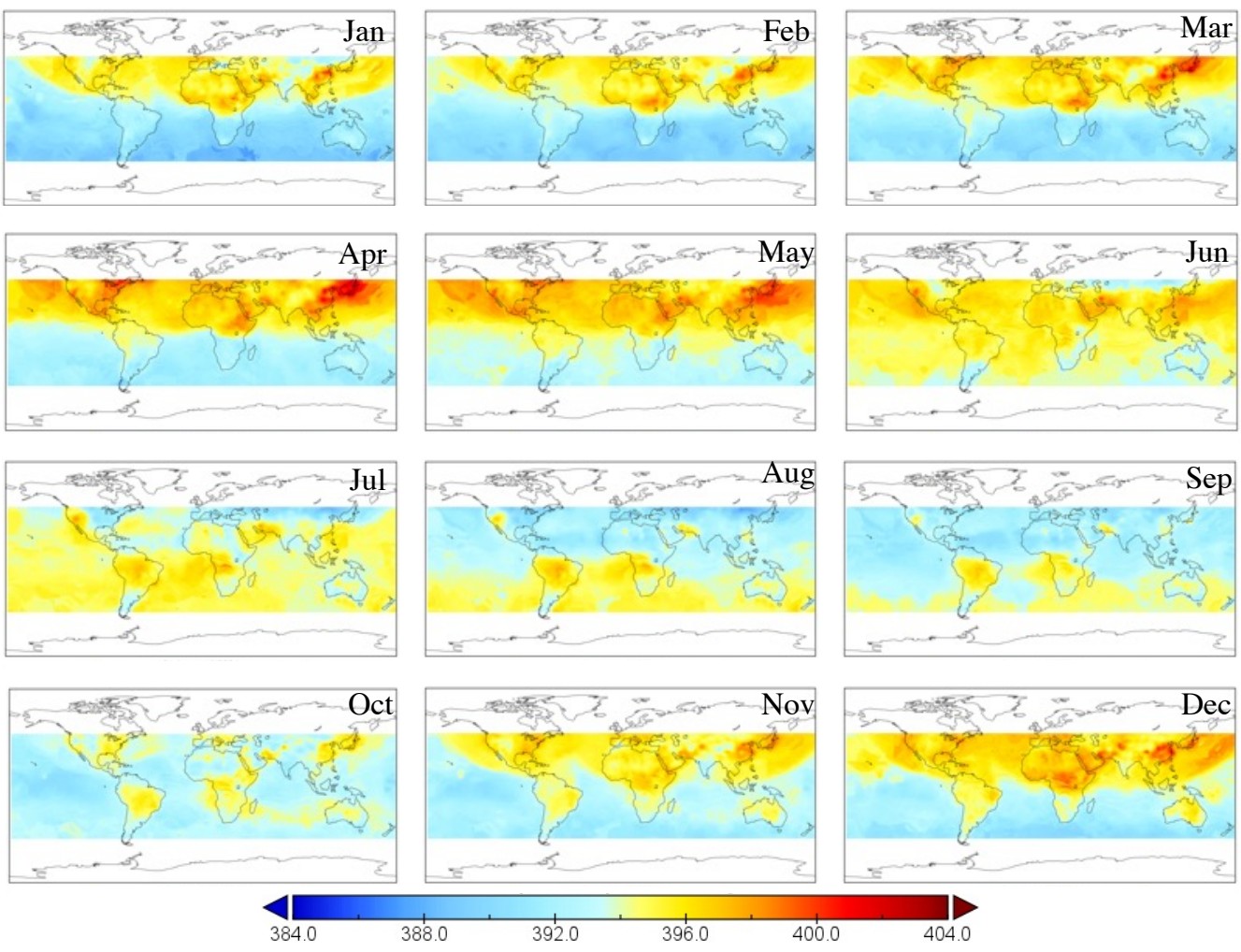

**Figure 10.** Product data from Our Algorithm in 2013. This CDC dataset covers approximately from 55°N to 55°S, with a spatial resolution of 0.25°.





**Figure 11.** Product data from Our Algorithm in 2014. This CDC dataset covers approximately from 55°N to 55°S, with a spatial resolution of 0.25°.





**Figure 12.** Product data from Our Algorithm in 2015. This CDC dataset covers approximately from 55°N to 55°S, with a
spatial resolution of 0.25°.


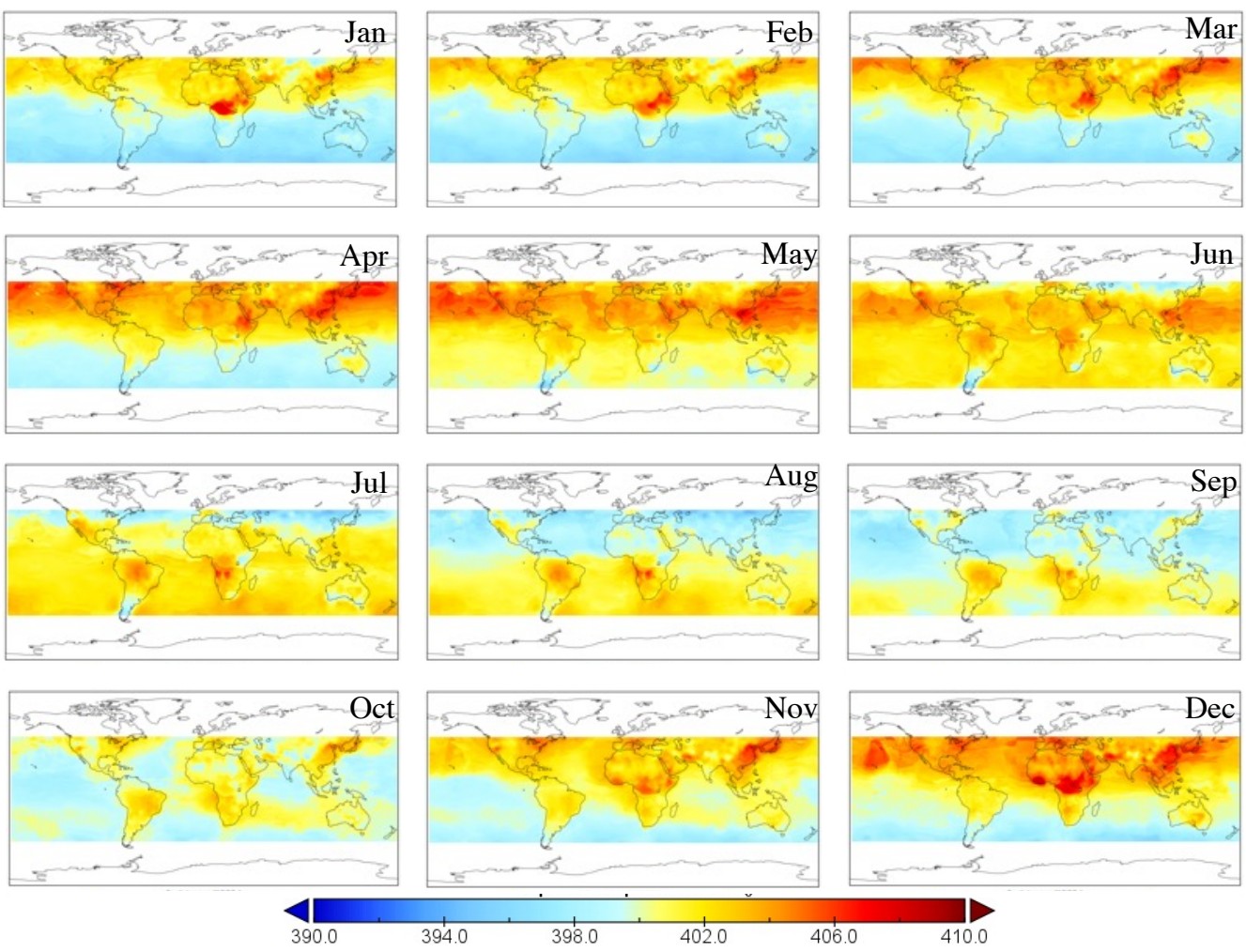

**Figure 13.** Product data from Our Algorithm in 2016. This CDC dataset covers approximately from 55°N to 55°S, with a spatial resolution of 0.25°.





**Figure 14.** Product data from Our Algorithm in 2017. This CDC dataset covers approximately from 55°N to 55°S, with a spatial resolution of 0.25°.



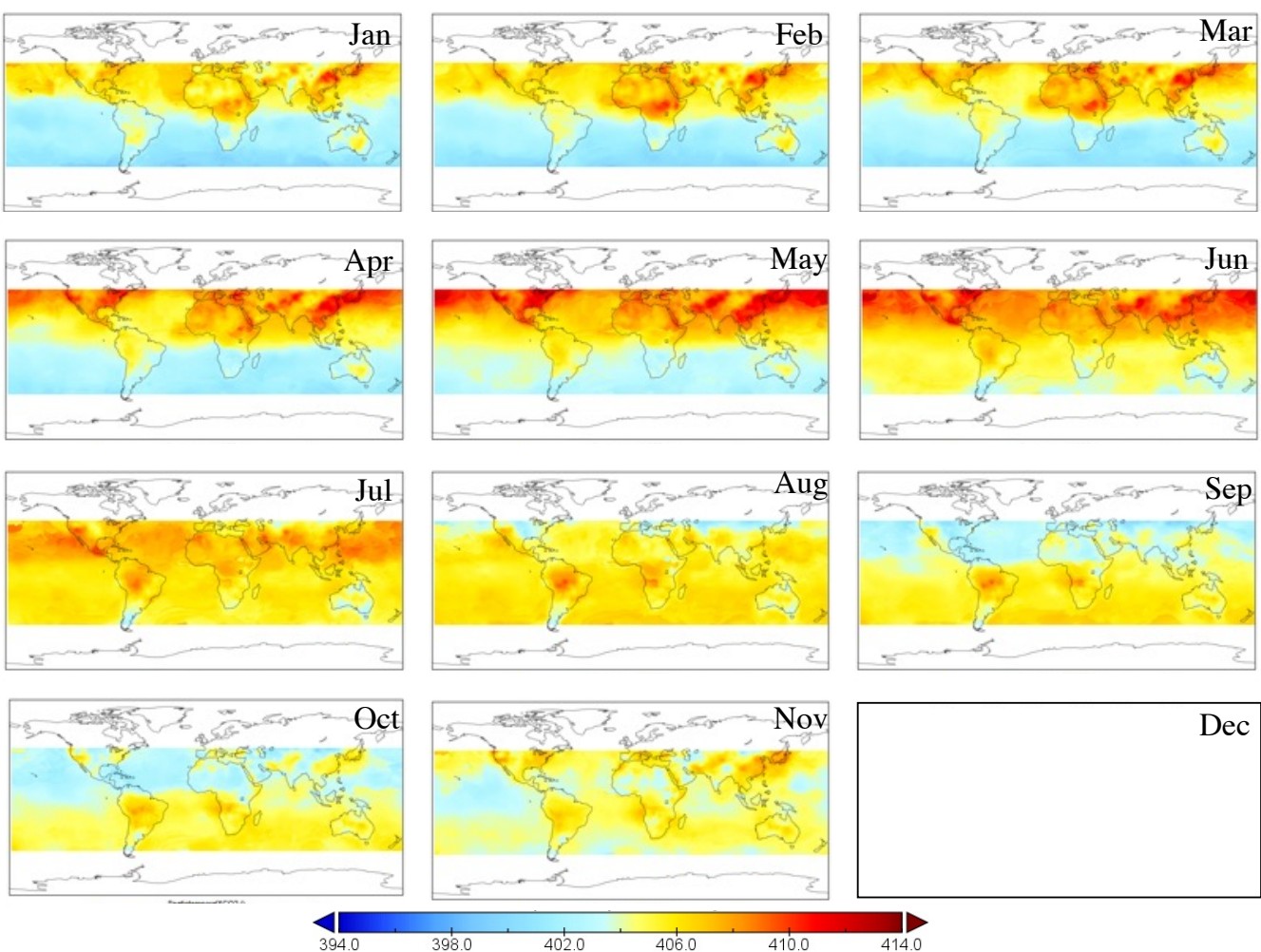

**Figure 15.** Product data from Our Algorithm in 2018. This CDC dataset covers approximately from 55°N to 55°S, with a
spatial resolution of 0.25°.




**Figure 16.** Product data from Our Algorithm in 2019. This CDC dataset covers approximately from 55°N to 55°S, with a spatial resolution of 0.25°.




**Figure 17.** Product data from Our Algorithm in 2020. This CDC dataset covers approximately from 55°N to 55°S, with a
spatial resolution of 0.25°.