# Peer review of "Carbon dioxide cover: carbon dioxide column concentration seamlessly distributed globally during 2009–2020"

_Earth System Science Data, 2022_

## Referee Comment (RC2)

Review of essd-2022-215

Zhang et al., "Carbon dioxide cover: carbon dioxide column concentration seamlessly distributed globally during 2009–2020"

This paper describes a gap-fill method for discrete $XCO_2$ data from the GOSAT satellite data product, and the resulting dataset, the Carbon Dioxide Coverage (CDC). The CDC contains monthly $XCO_2$ data with a spatial resolution of 0.25° × 0.25° grid box, globally for the period 2009–2020. The authors further make efforts to validate the dataset with other $XCO_2$ data from TCCON and OCO-2. The CDC dataset has already been made available on the "figshare" website and is downloadable with a readme file. Such a statistical mapping dataset makes it possible to understand global $XCO_2$ features from discrete satellite data. However, I found some parts of the manuscript difficult to follow, and I think the authors need to elaborate more on the data, the method used, and on the method of evaluation. As the focus of the manuscript seems to be on the technical description of the method, I feel that this manuscript with major revision may be more suitable for a technical journal or a journal on data modeling or analysis.

General comments:

1.  The authors should clarify the reason more why they used $XCO_2$ from the GOSAT FTS SWIR Level 3 (L3) product and not the FTS SWIR Level 2 (L2) product, to create the CDC dataset. The GOSAT L3 product is processed dataset of $XCO_2$ in GOSAT L2 product that filled the gaps in the L2 $XCO_2$ data with a Kriging method. The L3 product contains monthly-averaged $XCO_2$ values on a 2.5 degree latitude × 2.5 degree longitude grid. The L3 product covers all grids on Earth, but any mesh with no L2 data within a circle of 500-km radius were masked in the product to remove estimated data with extremely high minimum mean square errors (Watanabe et al., 2015). So, I am wondering why the authors chose GOSAT L3 product in which a Kriging method was already applied. Usually, these kinds of statistical gap-fill methods are applied to $XCO_2$ in the GOSAT or OCO-2 L2 data products which contains retrieved $XCO_2$ at each satellite observation point from spectral data in the Level 1B product (e.g. Hammerling et al., 2012; Liu et al., 2012; Watanabe et al., 2015). It would be more effective/impressive to show the usefulness of the proposed method in this manuscript if the authors applied the method to L2 $XCO_2$ data as the L2 $XCO_2$ data is more sparse

than L3 XCO$_2$ data.

2. The authors emphasis that the proposed method with the EBK methods for spatial prediction and time variation of XCO$_2$ improve the dataset accuracy effectively (e.g. Abstract, L65, 90, 146). However, it seems inadequate to conclude so, only from written results of the evaluations in the current manuscript which shows the results from the CDC dataset only. For more detailed evaluation for the authors' intended conclusion, the CDC dataset should be compared to XCO$_2$ dataset with previous methods such as the ordinary Kriging method as well as the original GOSAT L3 XCO$_2$ against TCCON observations.

3. Overall, the manuscript is somewhat vague and suffers from numerous incorrect descriptions. The presentation style could certainly be improved through corrections and elaborating the procedure more so that readers can understand the data processing procedure used to create the CDC dataset. Please see specific comments below for detail.

Specific comments:

L1: It seems to me that the title does not summarize the contents. One cannot guess the contents is on observations, model simulations, data analysis or dataset on CO2 from the title. The title should be more specific.

L11, 43: XCO$_2$ is the column-averaged dry-air mole fraction of CO$_2$ and is not equal to simple CO$_2$ concentration. Please define XCO$_2$ correctly.

L13, 35, 44 etc.: I cannot understand what the authors mean by "sniffing" in "carbon sniffing satellites" or "sniffer" in "sniffer satellites".

L13: Please replace "Gosat-2" by "GOSAT-2". A proper noun such as satellite names should be referred correctly.

L18, 67: Please clarify "the raw GOSAT data". In the later section, the authors write that they used GOSAT L3 product which is not "raw" data but processed data.

L29: I cannot understand what the authors mean by "respectively" in this sentence.

L32: Did the authors use "R" here, not $R^2$? Please clarify why they use R only here, while $R^2$ is used elsewhere.

L63: Please clarify "the prior time profile information of $XCO_2$". Is this time series of observed XCO2 or model simulated XCO2 or something else that is prepared prior to the analysis?

L65: What are "the original data" and "the prior information" exactly in this case?

L72: I cannot find in the method section how the "populated data" is used in this study. Please describe the populated data in the methods section.

L84: The authors write the spatial coverage of CDC is "from 50°S to 50°N" in Line 76 and "55°S to 55°N" in Line 84. Which is correct for the CDC dataset?

L87: What is "the model" here? Please define.

L89, 97: Replace "Tansat" by "TanSat".

L98, 105: I think that "GOSAT_FTS_L3_V2.95" and "OCO-2_L2_Lite_FP9r" are not proper product names. The authors need to clarify exactly the dataset they used; the product name, version, data provider etc.

L99: I could not find the comparison results of "0.56 ppm" in Watanabe et al., 2015. "0.56 ppm" in Noel et al., 2021 is for retrieved XCO2 by the ACOS FOCAL algorithm, but not for Level 3 product used here. The authors should read papers carefully and refer correctly.

L100: Do the authors use GOSAT-2 product as well?

L105: Related to my general comment #1, it is unclear why the authors discuss Level 3 and Level 2 products in same manner as the purposes and properties of the products

are different. The product the authors used in this study is GOSAT Level 3 product which is processed $XCO_2$ data with fixed grids in monthly interval processed by a geostatistical mapping approach (a Kriging) based on Level 2 product. While the "OCO-2_L2_Lite_F99r" is Level 2 product, it is natural that data locations are shifted along the satellite orbit. If the GOSAT L3 data used here is from NIES, Japan, their GOSAT L2 product is also downloadable from their website.

L109: The GOSAT TANSO-FTS sensor has seven channels: six for SWIR three bands and one for TIR, and GOSAT also carries TANSO-CAI sensor (JAXA/NIES/MoE, "GOSAT/IBUKI Data Users Handbook", 2011). So, "the six data channels of the sensor" may be replaced by "six of the seven channels of the TANSO-FTS sensor".

L110: The sentence is not correct. GOSAT collect data even in polar regions during winter. Just the spectral data in such regions are not used in $XCO_2$ retrieval processing for the SWIR Level 2 product.

L112: Please replace http by https.

L123-124, 125: The numbers, 0.9686, 1.3811, and 0.7, suddenly appear without any explanation about the evaluation method. Since this section is on "Materials and Methods", the results should be written in the results section.

L127: Was this analysis performed using a particular software program and/or libraries? Please give software information in the method section so it can be referred by other investigators.

L136: The authors did not mention the data provider of Level 3 data used in this study. It is supposed from the version "V2.95" that the authors use NIES L3 product. If so, then the product is already monthly-averaged. Please explain how the authors perform monthly-averaged calculations on the monthly L3 product.

L135-145: Since the $XCO_2$ variations are largely different over land and oceans, it is expected that semi-variograms may also differ. Did the author examine semi-variogram models in this aspect?

L136: Please clarify the method of downscaling from 2.5°degrees in NIES L3 product to 0.25° in this study.

L153-154: Monthly L3 data is used for the spatial interpolation with the EBK theory to derive the monthly XCO2 dataset (i.e. monthly to monthly). Are the resulted monthly $XCO_2$ time series in the 0.25° resolution actually anomalous among adjacent times? Then, it would be useful to include a sample figure to show how the time curve parameter library overcome this issue.

L155: What is "a time profile parameter library"? Is it a data matrix of parameters for Eq. (1) at each point or a kind of software library? Please clarify.

L156: Is "each point" in the sentence specifically the original L3 grid points (2.5° mesh) or the CDC dataset points (0.25° mesh)?

L160: The $CO_2$ trend is not perfectly linear for time scale of 2000-2020. Does the linear fit (a + b*t) show enough performance for this period? Have the authors checked the residuals from the fitted curves?

L160: Is the equation (1) correct? The 3rd and 5th terms, and the 4th and 6th terms are the same, respectively.

L163: I cannot follow the idea by the authors to use TCA theory to integrate spatial and temporal attributes. By using the EBK theory in Section 2.3.1, the authors already obtain monthly $XCO_2$ at 0.25° mesh based on monthly $XCO_2$ in L3 product. Then it would be possible that the author can simply calculate time series at each 0.25° mesh based on Eq. (1) to fill time-gap in the original L3 product though it has only a few missing values. Does TCA theory have some advantages?

L167: I cannot understand why only b and c in Eq. (1) are representative and considered here. Please clarify.

L177: Did the authors examine other method in transfer learning and compared the results?

L184: Does "the model predictions" mean the CDC dataset? What is "the input data"?

L185: The authors chose the coefficient of determination ($R^2$) and the root mean square error (RMSE) to evaluate the proposed algorithm to create the CDC dataset. However, the absolute biases of the dataset against the observation from TCCON or OCO-2 cannot be evaluated by $R^2$ and RMSE. For example, predicted $XCO_2$ in some TCCON sites like BU and HF in Figure 3 show apparent biases against TCCON observations. The bias calculation should be considered for the evaluation in addition to the two indicators.

L191: Are definitions of P and R for $R^2$ (Eq. (4)) correct? Usually, they are opposite, i.e. P for observation and R for predicted values.

L194: There are several months with no data in Figures 7 to 17. Since the authors already made temporal interpolation in sections 2.3.2 and 2.3.3, I expected that the CDC results have no missing data in temporal direction. Please explain this point.

L195: Table 3 appears before Tables 1 and 2. Please renumber the tables.

L197: Please write more detail about the method of comparison. TCCON observations are continuous observation in time, while GOSAT observations are made at about 13:00 local time. Hence $XCO_2$ in GOSAT L2 and L3 products are largely represent $XCO_2$ in 13:00 local time. Which time span in a day of TCCON observation did the authors use to make monthly averages? Also, spatially, how was the CDC dataset at 0.25 mesh compared to XCO2 at TCCON sites (e.g. use spatially interpolated values into TCCON site locations or just use the mesh value at where TCCON sites are located).

L199: Why did the authors choose numbers 0.95 for $R^2$ and 1.5 for RMSE for criteria?

L200: The sentence says "24 TCCON sites", while Table 1 has 23 TCCON sites. "24" might be "23"?

L203: It would be helpful to see the dataset features if the authors could provide figures of time series of $XCO_2$ from the CDC dataset, GOSAT L3 product, and TCCON observations at each TCCON sites.

L204: Li et al. (2022) produce 8-day $XCO_2$ dataset from OCO-2 L2 product and Zeng et al. (2014) used ACOS-GOSAT Level 2 XCO2 product. GOSAT L3 product used in this study is already monthly-mean and smoothed onto 2.5° grid and based on bias-corrected L2 product by TCCON observations. Thus, it is not possible to judge the proposed method, i.e. the CDC dataset, is more refined than the previous work. The authors need to consider various factors to compare their results with the previous work.

L210: What is "the passive inversion mode"? Please describe the word.

L215-216: Please describe the processing method in detail: what "bad data" is, the "statistics" the author used, spatial resolution for monthly-average, time-span for monthly-average etc.

L216, 218: Previously, the authors use $R^2$, but here R is used for the evaluation. What is the reason for changing the evaluation metrics?

L220: It seems that the slopes of the fitted lines in orange in Figure 4 do not fit to centers of high-density areas in red. Are the figures okay?

L221-222: I think only through this comparison, the accuracy and stability of the CDC dataset may not be justified.

L223: It is not clear why the authors conclude the deviations is due to "revisit period" of the satellites. Please clarify.

L224-225: The authors mention about GOSAT-2 in this section. Did the authors use GOSAT-2 data in addition to GOSAT L3 product?

L225: There are no "official algorithms" of OCO-2 and GOSAT-2. What do the authors mean by that?

L230: How many datasets with removed data did the authors prepare for the evaluation?

L240: Same question in L194. Doesn't the time interpolation used in this study cover the no data period in the original L3 product?

L243: The labels are useful for dataset users, but the title is misleading because evaluation of uncertainty itself is not mentioned in this section. By using Kriging methods, Hammerling et al. (2012) and Watanabe et al. (2015) and others estimated data uncertainties in each grid. Can the authors also estimate uncertainty of the CDC dataset?

L247: L3 product is not observations, but processed data.

L247, 249: Did the authors use GOSAT-2 product as well?

L252: The sentence says "uncertainty", but "uncertain" in Table3 and the CDC dataset (h5 files) on the website.

L253: The information on GOSAT L3 is missing in this section.

L258-259: These lines are about latitude range which is already mentioned elsewhere. Why did the authors write the same sentence again in this section?

L261-262: The last sentence about the compressed code seems to be on the CDC dataset. Maybe better to move the sentence in L257.

L265-278: The sentences are almost the same as the abstract. Do the authors have any other topics that they contend lastly?

L279: Again, the same question. Did the author use GOSAT-2 data too?

L288-289: The author TS is missing?

L288(References): Please check the format of the references. Some are not in ESSD style.

L445(Table 2): Which is "Nums" in Table 2: the number of original OCO-2 data or the

number of monthly-averaged data?

L445(Table 2): Is it right that the evaluation method is R, not $R^2$?

L470(Table 3): "Data type" column seems not well organized because "Matrix" and "Float"/"Int" are in different categories. "Scalar" may be paired with "Matrix". Scalar and Matrix can have data in Float or Integer format.

L485(Figure 1): It would be useful to show TCCON sites that are used in this study in different color in the figure because the CDC dataset does not cover high latitudes and some sites were not used for the evaluation.

L505(Figure 2): What are "the official website" and "Shp" in Figure 2? Please clarify.

---

## Author Comment (AC1)

Long-term and regional scale monitoring of $CO_2$ from space is important for understanding climate changes. Satellite can cover globally but clear sky ratios vary much region by region. Spatial-temporal technique are useful. However, I found several critical issues in this paper. I recommend resubmission.

(1) GOSAT sampling pattern and $CO_2$ density enhancement over large emission sources; The GOSAT sampling pattern consists of grid observation and target observation. The sampling pattern is not uniform. GOSAT is targeting global megacities which shows local enhancement. Over the ocean, GOSAT is tracking the specular reflection points of the sun, of which sampling pattern is not grid. Authors should describe in more detail how to use these data for analysis.

Dear reviewer, thank you for your kindly suggestions. The characteristics of GOSAT-1/2 data are regional $XCO_2$ monitoring, and also the function of target observation. Therefore, GOSAT satellite data can be used to calculate carbon emissions from wildfires (Guo M et al., 2017), megacities (Kuze A et al., 2022;Shiomi K et al., 2022), and combined GOSAT-2 and OCO-2 data to calculate terrestrial carbon flux (Wang H et al., 2019). Fig. A1 shows a flowchart of A. Kuze's work (Kuze A et al., 2022;Shiomi K et al.,2022). This work investigates the relationship between the enhancement of near-surface $XCO_2$ (about 0-4 km) in a time series of dense target observations over megacities and the inverse of the simulated wind speed. This relationship is used to estimate surface $CO_2$ emissions. The averaged emission intensity for each city was estimated from linear regression slopes in six megacities (Beijing, New Delhi, New York City, Riyadh, Shanghai, and Tokyo). Therefore, the purpose of our work is to fill the data gaps captured by the GOSAT satellite, which may be due to clouds and aerosols, or some locations that cannot be observed by the satellites. With the obtained global high-density coverage data, emission calculations, carbon source and sink, and carbon cycle analysis can be developed more deeply based on GOSAT satellite data (Frankenberg C et al., 2011; Houweling S et al., 2015; Chevallier F et al., 2009; Wang J et al., 2020).

[Figure]

**Fig. A1.** Flowchart of this study. Critical data are the GOSAT XCO2 LT data products, wind speed from the HYSPLIT transport model, and the ODIAC inventory (Kuze A, 2022)

**References:**

Guo M, Li J, Xu J, et al. CO2 emissions from the 2010 Russian wildfires using GOSAT data[J]. Environmental pollution, 2017, 226: 60-68.

Kuze A, Nakamura Y, Oda T, et al. Examining partial-column density retrieval of lower-tropospheric CO2 from GOSAT target observations over global megacities[J]. Remote Sensing of Environment, 2022, 273: 112966.

Shiomi K, Kikuchi N, Suto H, et al. Gosat Partial Column Observation for Better Quantifying Urban CO 2 Flux[C]//IGARSS 2022-2022 IEEE International Geoscience and Remote Sensing Symposium. IEEE, 2022: 4350-4352.

Wang H, Jiang F, Wang J, et al. Terrestrial ecosystem carbon flux estimated using GOSAT and OCO-2 $XCO_2$ retrievals[J]. Atmospheric Chemistry and Physics, 2019, 19(18): 12067-12082.

Frankenberg C, Fisher J B, Worden J, et al. New global observations of the terrestrial carbon cycle from GOSAT: Patterns of plant fluorescence with gross primary productivity[J]. Geophysical Research Letters, 2011, 38(17).

Chevallier F, Maksyutov S, Bousquet P, et al. On the accuracy of the CO2 surface fluxes to be estimated from the GOSAT observations[J]. Geophysical Research Letters, 2009, 36(19).

Houweling S, Baker D, Basu S, et al. An intercomparison of inverse models for estimating sources and sinks of CO2 using GOSAT measurements[J]. Journal of Geophysical Research: Atmospheres, 2015, 120(10): 5253-5266.

Wang, J., Feng, L., Palmer, P.I. et al. Large Chinese land carbon sink estimated from atmospheric carbon dioxide data. Nature 586, 720–723 (2020). https://doi.org/10.1038/s41586-020-2849-9

(2) GOSAT data source;I do not understand why authors use the NIES level 3 products. Level 3 products are spatially interpolated already. As mentioned in (1), they are problematic. There are several Level 2 GOSAT products other than NIES such as ACOS, RemoTeC, University of Leicester, and JAXA. Why do authors use NIES products? There is no product defined as "official".

Dear reviewer, thank you for your kindly suggestions. The Level 3 data products are generated from Short Wavelength Infra-Red (SWIR) data observed by Thermal and Near-infrared Sensor for carbon Observation-Fourier Transform Spectrometer (TANSO-FTS) onboard Greenhouse gases Observing Satellite (GOSAT) (hereafter abbreviated as FTS SWIR L3 data products) and that are distributed by the National Institute for Environmental Studies, Japan. The Level 3 products from NIES are downloaded from the GOSAT project data center. And the GOSAT Project is a joint effort promoted by the Japan Aerospace Exploration Agency (JAXA), the National Institute for Environmental Studies (NIES), and the Ministry of the Environment (MOE). Therefore, we consider it to be the official product in the previous manuscript version. But this statement was not rigorous, and we modified the expression that the GOSAT product is the official product in the manuscript. And we added some introduction about GOSAT data, with the addition of " The Level 3 data products are generated from Short Wavelength Infra-Red data observed by Thermal and Near-infrared Sensor for carbon Observation-Fourier Transform Spectrometer (TANSO-FTS) onboard Greenhouse gases Observing Satellite (GOSAT) and that are distributed by the National Institute for Environmental Studies, Japan. The Level 3 products from NIES are downloaded from the GOSAT project data center. And the GOSAT Project is a joint effort promoted by the Japan Aerospace Exploration Agency (JAXA), the National Institute for Environmental Studies (NIES), and the Ministry of the Environment (MOE). "

We are also learning about other GOSAT L2 level products, such as ACOS, RemoTeC, University of Leicester, and JAXA, which are obtained based on the raw data from GOSAT satellite observations through the algorithms of different scientist teams. These works are very well done and have achieved good applications in terrestrial ecological carbon exchange, carbon emission calculation, carbon cycle, carbon sources and sinks (Frankenberg C et al., 2011; Houweling S et al., 2015; Chevallier F et al., 2009; Wang J et al., 2020; Jiang F et al., 2022; Liu J et al., 2014). However, the NIES level 3 products data was selected for several reasons:

1) The Level 3 data is processed based on the Level 2 data. Therefore, the Level 3 data inherits the quality of Level 2 data. And the Level 3 data fills the gaps of Level 2 data on the basis of retaining the quality of Level 2 original data. Besides, benefiting from the format of Level 3 data being fixed point data, the percentage of valid months in the algorithm can be greatly increased during the construction of the parameter library compared to the Level 2 GOSAT products. Within the pre-defined grid, there is a valid cumulative number of months within a year. If the number of valid cumulative months exceeds 10 for the current grid, the data from this grid will be involved in the subsequent construction of the time profile library to obtain a set of time profile parameters. Moreover, the curves that do not satisfy the pattern of $CO_2$ concentration change curves will be removed from the parameter curve library based on a priori knowledge. Therefore, Level 3 data will be more suitable for our algorithm compared to Level 2 data.

2) The NIES L3 data development and the GOSAT-1/2 satellites that were launched both came from. the Japanese government. Therefore, we concluded that the NIES L3 data would exist a good follow-up data maintenance expected from the Japanese government. Considering the continuous updating and maintenance of our dataset, the NIES L3 data can provide a strong guarantee for continuous updating.

3) The NIES L3 data is extremely easy to find and download from the Japanese GOSAT project data center (https://data2.gosat.nies.go.jp/GosatDataArchiveService/usr/auth/login (last access: 27-October-2022)).

   Therefore, for the above reasons, we prefer to select the data from the Japanese GOSAT project data center. Finally, some links are provided to obtain products based on the algorithms of ACOS, RemoTeC, University of Leicester.

1) An $XCO_2$ dataset developed based on the ACOS algorithm: (https://search.earthdata.nasa.gov/search/granules/collection-details?p=C1633158704-GES_DISC&pg[0][v]=f&pg[0][gsk]=-start_date&q=ACOS_L2S%207.3&tl=1666836734.099!3!!&lat=11.05508932556063&long=59.484375 (last access: 27-October-2022))

2) An $XCO_2$ dataset developed based on the RemoTeC algorithm: ( https://cds.climate.copernicus.eu/cdsapp#!/dataset/satellite-carbon-dioxide?tab=overview (last access: 27-October-2022))

3) An $XCO_2$ dataset developed from University of Leicester: (https://cds.climate.copernicus.eu/cdsapp#!/dataset/satellite-carbon-dioxide?tab=form (last access: 27-October-2022))

**References:**

Frankenberg C, Fisher J B, Worden J, et al. New global observations of the terrestrial carbon cycle from GOSAT: Patterns of plant fluorescence with gross primary productivity[J]. Geophysical Research Letters,

2011, 38(17).

Chevallier F, Maksyutov S, Bousquet P, et al. On the accuracy of the CO2 surface fluxes to be estimated from the GOSAT observations[J]. Geophysical Research Letters, 2009, 36(19).

Houweling S, Baker D, Basu S, et al. An intercomparison of inverse models for estimating sources and sinks of CO2 using GOSAT measurements[J]. Journal of Geophysical Research: Atmospheres, 2015, 120(10): 5253-5266.

Wang, J., Feng, L., Palmer, P.I. et al. Large Chinese land carbon sink estimated from atmospheric carbon dioxide data. Nature 586, 720–723 (2020). https://doi.org/10.1038/s41586-020-2849-9

Jiang F, Ju W, He W, et al. A 10-year global monthly averaged terrestrial net ecosystem exchange dataset inferred from the ACOS GOSAT v9 XCO 2 retrievals (GCAS2021) [J]. Earth System Science Data, 2022, 14(7): 3013-3037.

Liu J, Bowman K W, Lee M, et al. Carbon monitoring system flux estimation and attribution: impact of ACOS-GOSAT XCO2 sampling on the inference of terrestrial biospheric sources and sinks[J]. Tellus B: Chemical and Physical Meteorology, 2014, 66(1): 22486.

**<Specific Comments>**

(1) 2.2 validation data TCCON; When the authors use the muti-year data in TCCON comparison, the coefficient of determination becomes too good. The annual growth of global $CO_2$ density should be removed for the analysis. The deviation and bias of matched up data should be presented.

Dear reviewer, thank you for your kindly suggestions. This coefficient of determination is obtained by matching the product data to the TCCON value in space and time. Thus, we obtained the coefficient of determination values in Table 1 at each TCCON site. This coefficient of determination value was obtained normally.

Our work is to produce the monthly-averaged $XCO_2$ product set in this manuscript. For the analysis of interannual $XCO_2$, we produced scatter error plots for different years at each TCCON site. Moreover, the annual-averaged $XCO_2$ of TCCON stations and the predicted data annual-averaged $XCO_2$ in different years are plotted in the station subplots in Fig 2.2-1. And the error bars represent the evaluated RMSE from the TCCON site and the predicted data within different years. Table 2.2-1. represents the annual- averaged errors for different TCCON stations for many years with the inclusion of the indicators $R^2$ and RMSE. The multi-year $XCO_2$ evaluation index $R^2$ is above 0.99 for all 23 sites, with the RMSE evaluation index interval ranging from 0.088 to 0.957 ppm. In particular, the annual-averaged $XCO_2$ evaluation index $R^2$ for all sites was 0.999 and the RMSE evaluation index was 0.283 ppm. The deviations and biases of the data matching have been presented and added to Table. 1 in the latest manuscript.

**Table 2.2-1.** Geographic locations of TCCON sites used for validation and the statistics used to compare predicted annual-averaged $XCO_2$ and TCCON $XCO_2$ observations. The " - " represents the number of years for the site is less than 3.

| Tccon sites (Site abbreviations) | $R^2$ | RMSE (ppm) |
| --- | --- | --- |
| Jet Propulsion Laboratory (JC) | 0.988 | 0.957 |
| Caltech (CI) | 0.999 | 0.232 |
| Edwards (DF) | 0.999 | 0.174 |
| Four Corners (FC) | - | - |
| Lamont (OC) | 0.998 | 0.356 |

| | | | | | |
|---|---|---|---|---|---|
| Park Falls (PA) | 0.996 | | 0.568 | | |
| Manaus (MA) | - | | - | | |
| Izana (IZ) | 0.998 | | 0.387 | | |
| Ascension Island (AE) | 0.986 | | 0.568 | | |
| Orléans (OR) | 0.999 | | 0.329 | | |
| Zugspitze (ZS) | 0.994 | | 0.529 | | |
| Garmisch (GM) | 0.997 | | 0.503 | | |
| Nicosia (NI) | - | | - | | |
| Réunion Island (RA) | 0.994 | | 0.549 | | |
| Hefei (HF) | - | | - | | |
| Burgos (BU) | 0.994 | | 0.312 | | |
| Anmeyondo (AN) | 0.999 | | 0.088 | | |
| Saga (JS) | 0.995 | | 0.524 | | |
| Edwards (DB) | 0.997 | | 0.436 | | |
| Tsukuba (TK) | 0.987 | | 0.771 | | |
| Rikubetsu (RJ) | 0.991 | | 0.478 | | |
| Wollongong (WG) | 0.997 | | 0.497 | | |
| Lauder01&02&03 (LL) | 0.996 | | 0.530 | | |
| AllSites | 0.999 | | 0.283 | | |

**Table 1.** Geographic locations of TCCON sites used for validation and the statistics used to compare predicted $XCO_2$ and TCCON $XCO_2$ observations.

| Tccon sites (Site abbreviations) | Longitude | Latitude | $R^2$ | RMSE (ppm) | MEAN (ppm) |
|---|---|---|---|---|---|
| Jet Propulsion Laboratory (JC) | -118.18 | 34.20 | 0.98** | 1.07 | -0.89 |
| Caltech (CI) | -118.13 | 34.14 | 0.97** | 0.95 | -1.21 |
| Edwards (DF) | -117.88 | 34.96 | 0.98** | 0.82 | 0.72 |
| Four Corners (FC) | -108.48 | 36.80 | 0.96** | 0.31 | -0.75 |
| Lamont (OC) | -97.49 | 36.60 | 0.98** | 1.04 | -1.79 |
| Park Falls (PA) | -90.27 | 45.94 | 0.98** | 1.24 | -0.62 |
| Manaus (MA) | -60.60 | -3.21 | 0.88** | 0.64 | -1.53 |
| Izana (IZ) | -16.48 | 28.30 | 0.98** | 1.18 | -0.96 |
| Ascension Island (AE) | -14.33 | -7.92 | 0.94** | 0.93 | -0.84 |
| Orléans (OR) | 2.11 | 47.97 | 0.99** | 0.95 | 0.13 |
| Zugspitze (ZS) | 10.98 | 47.42 | 0.92** | 1.52 | -0.40 |
| Garmisch (GM) | 11.06 | 47.48 | 0.98** | 1.05 | 0.36 |
| Nicosia (NI) | 33.38 | 35.14 | 0.93** | 0.73 | -1.38 |
| Réunion Island (RA) | 55.49 | -20.90 | 0.96** | 1.23 | -1.33 |
| Hefei (HF) | 117.17 | 31.90 | 0.87** | 1.51 | -1.83 |
| Burgos (BU) | 120.65 | 18.53 | 0.89** | 1.01 | -1.58 |
| Anmeyondo (AN) | 120.65 | 36.54 | 0.90** | 1.20 | -0.58 |
| Saga (JS) | 130.29 | 33.24 | 0.97** | 1.26 | -1.14 |
| Edwards (DB) | 130.89 | -12.43 | 0.99** | 0.75 | -1.03 |
| Tsukuba (TK) | 140.12 | 36.05 | 0.91** | 1.89 | 0.48 |

| | | | | | |
|---|---|---|---|---|---|
| Rikubetsu (RJ) | 143.77 | 43.46 | 0.95** | 1.17 | 0.19 |
| Wollongong (WG) | 150.88 | -34.41 | 0.99** | 0.82 | -0.58 |
| Lauder01&02&03 (LL) | 169.68 | -45.04 | 0.97** | 1.44 | -0.70 |
| All sites | - | - | 0.97** | 1.38 | -0.64 |

[Figure]

**Figure 2.2-1** Scatter error plot of annual-averaged $CO_2$ concentration. AYPXCO$_2$ represents the predicted annual-averaged $CO_2$ concentration, AYTXCO$_2$ represents the annual-averaged $CO_2$ concentration at the TCCON site.

(2) 2.2 Validation data: OCO-2 Level 2 product; The version of the OCO-2 level 2 products should be described. Older OCO-2 products have topography dependent bias. The difference in footprints of GOSAT and OCO-2 creates errors.

A. The version of the OCO-2 level 2 products should be described. Older OCO-2 products have topography dependent bias.

Dear reviewer, thank you for your kindly suggestions. OCO-2_L2_Lite_FP9r (DOI: 10.5067/W8QGIYNKS3JC) is the used version of the OCO-2 data in the previous manuscript. This version has a time interval from 201409 to 202001. However, the National Aeronautics and Space Administration (NASA) team has provided the OCO-2_L2_Lite_FP10r version (DOI: 10.5067/E4E140XDMPO2) now, which mainly addresses the topography-dependent bias as well as the lack of data coverage in 2020.

By update the OCO-2_L2_Lite_FP10r version of the OCO-2 data, this revision reduces the impact of topography-dependent bias in the OCO-2 data and increases the amount of comparative data available in 2020. Therefore, we re-compared the results of the interpolated data with the OCO-2 data in the latest manuscript. The updated results are presented in Figure 4 and Table 2 in the following.

[Figure]

**Figure 4.** Density scatter plots of predicted $XCO_2$ and observed one from OCO-2.

**Table 2**. Statistics for predicted monthly averaged $XCO_2$ and OCO-2 monthly averaged $XCO_2$ observations.

| Year | $R$ | Nums |
|------|-----|------|
| 2014 | 0.38** | 131463 |
| 2015 | 0.77** | 594519 |
| 2016 | 0.76** | 791634 |
| 2017 | 0.80** | 641891 |
| 2018 | 0.70** | 706870 |

| 2019 | 0.76** | 785660 |
| 2020 | 0.75** | 756116 |

** At the 0.01 level (two-tailed), the correlation is significant.

**Here we provide a description of the CO-2_L2_Lite_FP10r data from the Nasa team.**

In early 2021, the OCO Team identified an issue with OCO-2 level 2 products processed since January 28, 2020. The Ancillary Geometric Product (AGAP) file, a static file used in OCO-2 Geolocation processing, was inadvertently replaced with an obsolete version. This AGAP file included a ~300 m pointing error. As a result, all OCO-2 Level 2, version 10r, data files for the period January 28 - December 31, 2020, were corrected and replaced. The replacement process was completed by the end of June, 2021. The significance of this error has been described in Kiel et al. (2019; doi:10.5194/amt-12-2241-2019).

**Here, we have provided links to OCO-2 data for both versions.**

OCO-2_L2_Lite_FP9r: (https://disc.gsfc.nasa.gov/datasets/OCO2_L2_Lite_FP_9r/summary (last access: 27-October-2022))

OCO-2_L2_Lite_FP10r:(https://search.earthdata.nasa.gov/search/granules/collection-details?p=C1685783927-GES_DISC&pg[0][v]=f&pg[0][gsk]=-start_date&q=OCO2_L2_Lite_FP (last access: 27-October-2022))

B. The difference in footprints of GOSAT and OCO-2 creates errors.

Dear reviewer, thank you for your kindly suggestions. TANSO-FTS-2 is the same as the Fourier Transform spectrometer on the first GOSAT satellite. And the Instant Field of View (IFOV) is 15.8 mrad, which is the same as GOSAT, but the footprint size is smaller - from 10.5 km to 9.7 km - due to the lower altitude of GOSAT-2. Besides, the dimension of 0.25 degree can be used for most requirements. Therefore, we choose 0.25 degree as the size of product data. And Our product data is developed based on mid- and low- latitude grid data. Firstly, the GOSAT data within each grid are averaged, and then subsequent processing is based on the algorithmic framework. Besides, when we did the validation comparison with the OCO-2 data, we directly averaged the OCO-2 data in the corresponding grid, which may contain multiple values of OCO-2. Therefore, the errors caused by the GOSAT data and OCO-2 footprints can be neglected in our paper.

**<Technical Corrections>**

(1) Page 15 Table 1; Is the unit of RMSE ppm?

Dear reviewer, thank you for your kindly suggestions. The unit of RMSE is ppm, And we have modified the questions you extracted in the latest manuscript. The modified Table 1 is shown in the following.

**Table 1.** Geographic locations of TCCON sites used for validation and the statistics used to compare predicted $XCO_2$ and TCCON $XCO_2$ observations.

| Tccon sites (Site abbreviations) | Longitude | Latitude | $R^2$ | RMSE (ppm) | MEAN (ppm) |
|---|---|---|---|---|---|
| Jet Propulsion Laboratory (JC) | -118.18 | 34.20 | 0.98** | 1.07 | -0.89 |
| Caltech (CI) | -118.13 | 34.14 | 0.97** | 0.95 | -1.21 |

| | | | | | |
|---|---|---|---|---|---|
| Edwards (DF) | -117.88 | 34.96 | 0.98** | 0.82 | 0.72 |
| Four Corners (FC) | -108.48 | 36.80 | 0.96** | 0.31 | -0.75 |
| Lamont (OC) | -97.49 | 36.60 | 0.98** | 1.04 | -1.79 |
| Park Falls (PA) | -90.27 | 45.94 | 0.98** | 1.24 | -0.62 |
| Manaus (MA) | -60.60 | -3.21 | 0.88** | 0.64 | -1.53 |
| Izana (IZ) | -16.48 | 28.30 | 0.98** | 1.18 | -0.96 |
| Ascension Island (AE) | -14.33 | -7.92 | 0.94** | 0.93 | -0.84 |
| Orléans (OR) | 2.11 | 47.97 | 0.99** | 0.95 | 0.13 |
| Zugspitze (ZS) | 10.98 | 47.42 | 0.92** | 1.52 | -0.40 |
| Garmisch (GM) | 11.06 | 47.48 | 0.98** | 1.05 | 0.36 |
| Nicosia (NI) | 33.38 | 35.14 | 0.93** | 0.73 | -1.38 |
| Réunion Island (RA) | 55.49 | -20.90 | 0.96** | 1.23 | -1.33 |
| Hefei (HF) | 117.17 | 31.90 | 0.87** | 1.51 | -1.83 |
| Burgos (BU) | 120.65 | 18.53 | 0.89** | 1.01 | -1.58 |
| Anmeyondo (AN) | 120.65 | 36.54 | 0.90** | 1.20 | -0.58 |
| Saga (JS) | 130.29 | 33.24 | 0.97** | 1.26 | -1.14 |
| Edwards (DB) | 130.89 | -12.43 | 0.99** | 0.75 | -1.03 |
| Tsukuba (TK) | 140.12 | 36.05 | 0.91** | 1.89 | 0.48 |
| Rikubetsu (RJ) | 143.77 | 43.46 | 0.95** | 1.17 | 0.19 |
| Wollongong (WG) | 150.88 | -34.41 | 0.99** | 0.82 | -0.58 |
| Lauder01&02&03 (LL) | 169.68 | -45.04 | 0.97** | 1.44 | -0.70 |
| All sites | - | - | 0.97** | 1.38 | -0.64 |

** At the 0.01 level (two-tailed), the correlation is significant.

(2) page 19, Figure 2; The branching of left and right should be described.

Dear reviewer, thank you for your kindly suggestions. The framework in Figure 2 depicts the general methodology. The workflow of the left branch of Figure 2 is to construct a time-curve parameter library based on the input data, and the workflow of the right branch of Figure 2 is to fill the gaps in the original data on the spatial attributes.

First, section 2.3.2 in the manuscript is visualized in the left branch of Figure2.We pre-divide the grid and then determine whether the input data within each sub-grid is valid input data based on the quality label of the input data. If the cumulative number of valid input data of the grid exceeds 10 in a period (12 consecutive months). Then this grid is a valid month grid and this grid can be used to build the time curve library parameters. The construction of time profile library parameters refers to the extraction of time profile parameters for the valid month grid combined with Equation 1. Finally, the time curve parameters that do not satisfy the requirements are removed based on the a priori knowledge of the $CO_2$ concentration change pattern. And the time profile parameters that meet the requirements are input to the time-profile parameter library.

Second, section 2.3.1 in the manuscript is visualized in the right branch of Figure 2. $XCO_2$ information gaps are filled based on the input data for each month under the perspective of spatial information. Then global mid- and low-latitude projections can be obtained for each month.

Lastly, we incorporated temporal information into the previously acquired surface interpolation data through the proposed transfer component analysis (TCA) theory.

[Figure]

**Figure 2**. Framework of the proposed methodology

---

## Author Comment (AC2)

In this paper, a spatiotemporal interpolation method is developed, and a data product with full spatiotemporal coverage is generated by using the $XCO_2$ data of GOSAT. I'm not particularly aware of the article category for ESSD, but compared to other research papers in ESSD, it's more suitable for technical description articles, at least at this stage. The article has some obvious scientific errors and inappropriate knowledge descriptions, the following points should be considered to improve the quality of the article, especially some major errors. As community comments, I believe these comments will increase the understanding of carbon monitoring satellite data assimilation and improve this research.

1. Although we are very concerned about the carbon cycle and the spatiotemporal distribution of $CO_2$, for atmospheric inversion models, sparse data observations are sufficient to obtain carbon fluxes. **NOTE I'm not denying that we don't need a spatially seamlessly $CO_2$ distribution,** but the introduction should explain why we need a spatiotemporally seamlessly $CO_2$, such as calculating global averages, analyzing seasonal changes.

Dear reviewer, thank you for your kindly suggestions. We have checked the description about carbon fluxes in the manuscript and made some modifications. The following shows the revised description in the latest manuscript:

Obtaining highly accurate and high-resolution spatiotemporal maps of $XCO_2$ distributions is essential for promoting the study of the carbon cycle, carbon sources, carbon sinks, carbon neutralization, and carbon emissions assessed by top-down theory. In addition, based on seamless $XCO_2$ data, relevant studies can be carried out, for example, to analyze global seasonal or annual changes (Zhang M et al., 2022), to calculate carbon emissions from wildfires (Guo M et al., 2017) or megacities (Kuze A et al., 2022; Shiomi K et al., 2022), to calculate terrestrial carbon fluxes (Wang H et al., 2019) by combining GOSAT-2 and OCO-2 data.

**References:**

Frankenberg C, Fisher J B, Worden J, et al. New global observations of the terrestrial carbon cycle from. GOSAT: Patterns of plant fluorescence with gross primary productivity[J]. Geophysical Research Letters, 2011, 38(17).

Guo M, Li J, Xu J, et al. CO2 emissions from the 2010 Russian wildfires using GOSAT data[J]. Environmental pollution, 2017, 226: 60-68.

Kuze A, Nakamura Y, Oda T, et al. Examining partial-column density retrieval of lower-tropospheric. CO2 from GOSAT target observations over global megacities[J]. Remote Sensing of Environment, 2022, 273: 112966.

Shiomi K, Kikuchi N, Suto H, et al. Gosat Partial Column Observation for Better Quantifying Urban CO. 2 Flux[C]//IGARSS 2022-2022 IEEE International Geoscience and Remote Sensing Symposium. IEEE, 2022: 4350-4352.

Wang H, Jiang F, Wang J, et al. Terrestrial ecosystem carbon flux estimated using GOSAT and OCO-2. $XCO_2$ retrievals[J]. Atmospheric Chemistry and Physics, 2019, 19(18): 12067-12082.

Zhang M, Liu G. Mapping contiguous $XCO_2$ by machine learning and analyzing the spatio-temporal. variation in China from 2003 to 2019[J]. Science of The Total Environment, 2022: 159588.

2. P6. Line 160. **Equation 1 looks very strange and seems wrong.** What is the difference between coefficients c and e? What is the difference between d and g? The result needs to be checked carefully and even recalculated. In addition, adopting this method to construct time series would lead to significant

drawbacks. I believe the authors may not understand the $CO_2$ growth rate, please see papers such as Buchwitz. et.al, 2018 and A. Chatterjee et.al, 2017 to understand the significance of CGR in reflecting vegetation, climate, etc. The method of presetting a function to fit will not be able to capture the real change in the $CO_2$ growth rate, and the function will be directly known after derivation. One of the reasons we want seamless data is to better calculate the global average, and thus the growth rate, that is, the net flux. Fitting with a fixed function would loss this.

Dear reviewer, thank you for your kindly suggestions. The corrected Equation 1, which was misrepresented in the previous manuscript (the code that has been uploaded in the repository is the correct representation), is shown below. The role of parameter $c$, parameter $e$, parameter $d$ and parameter $g$ are to adjust the month-to-month variation within the year (Fu P et al., 2019) the differences between them are the different terms that are adjusted.

$$F(t) = a + b * t + c * \cos\left(\frac{2\pi t}{f}\right) + d * \sin\left(\frac{2\pi t}{f}\right) + e * \cos\left(\frac{4\pi t}{f}\right) + g * \sin\left(\frac{4\pi t}{f}\right), \tag{1}$$

where $a$ refers to the yearly averaged $XCO_2$; $c$, $d$, $e$, and $g$ are the coefficients of the seasonal component; $b$ is the coefficient of the interannual component; $f$ is the sampling frequency ($f$ = 12 for a year); and $t$ is the sampling interval.

**References:**
P. Fu, Y. Xie, C. E. Moore, S. W. Myint, and C. J. Bernacchi, "A comparative analysis of anthropogenic CO2 emissions at city level using OCO2 observations: A global perspective," Earth's Future, vol. 7, no. 9, pp. 1058–1070, Sep. 2019

3. In the abstract and the introduction, the authors claim that the amount of $XCO_2$ data is mainly affected by factors such as clouds and aerosols. This is wrong, it's actually a swath issue. The design of the width is related to the fluctuation amplitude of atmospheric $CO_2$ and optical inversion factors. Authors are advised to read the relevant literature and correct the description in this section. While this is not that important for this article, the readers need to understand the real background for this work.

Dear reviewer, thank you for your kindly suggestions. The description of the pointed-out error has been modified in the abstract and introduction. The modified expression is "However, the discrete satellite data provided by GOSAT-2, OCO-2, and OCO-3 have data voids and relatively low efficiency because of narrow swaths and reduced sampling densities due to optically thick clouds and aerosols (Hu Y et al., 2021; Zhang M et al., 2022)."

**References:**
Hu Y, Shi Y. Estimating $CO_2$ emissions from large scale coal-fired power plants using OCO-2 observations and emission inventories[J]. Atmosphere, 2021, 12(7): 811.
Zhang M, Liu G. Mapping contiguous $XCO_2$ by machine learning and analyzing the spatio-temporal variation in China from 2003 to 2019[J]. Science of The Total Environment, 2022: 159588.

4. P3, L65-83. This section overlaps with method descriptions in subsequent part, and it is not appropriate to introduce too much about the methods of this study in the introduction.

Dear reviewer, thank you for your kindly suggestions. The contents of L65-83 were removed from the original manuscript.

5. P4, L98 "...the accuracy of the comparison between the GOSAT data product and the TCCON site was 0.56 ppm", it is not appropriate to use the **"accuracy"** word, it should be stated, such as standard deviation, bias, etc.

Dear reviewer, thank you for your kindly suggestions. By checking the literature (Noël S et al., 2021), this accuracy refers to an overall station-to-station bias. Therefore, the content of L98 has been modified and its modified content is as follows: "...the overall station-to-station bias between the GOSAT data product and the TCCON site was 0.56 ppm ".

**References:**

Noël, S., Reuter, M., Buchwitz, M., Borchardt, J., et al., XCO$_2$ retrieval for GOSAT and GOSAT-2 based on the FOCAL algorithm, Atmos. Meas. Tech., 14, 3837–3869, (2021)

6. P4, L100, "... three-day temporal resolution. The time resolution of GOSAT-2 satellite is 6 days..." is inappropriate. The correct description is the revisit cycle/repeat cycle. Please differentiate these concepts. (Temporal resolution, time resolution, repeat cycle,)

Dear reviewer, thank you for your kindly suggestions. The content of L100 has been modified and its modified content is as follows: "... and three-day revisit cycles. The revisit cycle of GOSAT-2 satellite is 6 days, IFOV is 9.7km. ".

7. P4, L105. However, the OCO-2_L2_Lite_FP9r provides data locations that are gradually shifted over time by satellite observations. This sentence is difficult to understand. I think you should express that the orbits of the sub-satellite points are evenly distributed? Illustration may be needed.

Dear reviewer, thank you for your kindly suggestions. We want to express that the spatial position of the satellite subsatellite point is gradually shifted under the same position of two adjacent revisit cycles. Fig. 7-1 demonstrates the spatial distribution of XCO$_2$ data. The data are from the OCO-2 satellite, which was collected on January 1, 2019 in Fig. 7-1a and Fig. 7-1b. Besides, the data are from the OCO-2 satellite, which was collected on January 17, 2019 in Fig. 7-1c and Fig. 7-1d. And the revisit period of OCO-2 satellite is 16 days. From the spatial position distribution of XCO$_2$ in Fig. d and Fig. b, the satellite orbit is gradually offset in order to collect more data.

[Figure]

Figure7-1 Spatial distribution of XCO₂ data from OCO-2 satellite.

8.P4, L112 column-averaged XCO₂. This is wrong. And the full name of XCO₂ is wrong, including the title, abstract and etc. I think it should be carefully checked the full text of the corresponding full scientific name.

Dear reviewer, thank you for your kindly suggestions. We have checked the full name of XCO₂ in the manuscript. And the full name of XCO₂ was modified to column-averaged dry-air mole fraction of CO₂ in the whole manuscript. And the new manuscript title is "Carbon dioxide cover dataset: XCO₂ seamlessly distributed globally during 2009-2020".

9. P4. L107, "fixed location" should be clearer. L109. the six data channels are wrong. TANSO-FTS is a 4–band interferometer.

Dear reviewer, thank you for your kindly suggestions. To make it easier to understand for readers, we have removed the description of L107 in the previous manuscript. "The GOSAT_L3 product provides a cumulative observation of a long time series in grid form." was added to the latest manuscript. TANSO-FTS is a four-band interferometer [1], with three bands located in the near-infrared and short-wave infrared [1], and with seven data channels [2-3]. In the latest manuscript, we have added the modified content "TANSO-FTS is a four-band interferometer, with three bands located in the near-infrared and short-wave infrared"。

**References:**
[1] https://space.oscar.wmo.int/instruments/view/tanso_fts, last access: 10-Nov-2022.

[2] https://www.gosat.nies.go.jp/en/about_%EF%BC%92_observe.html, last access: 10- Nov- 2022.

[3]https://seors.unfccc.int/applications/seors/attachments/get_attachment?code=645A2WJLB 852G36JR3WM5GECF1HMXXEP, last access: 10-Nov-2022.

10. P4., L123. Please correct for ***column-averaged abundances of CO₂*** expression. And the results showed that R2 was 0.9686, and RMSE was 1.3811. Please indicate the source.

Dear reviewer, thank you for your kindly suggestions. For your question, we have divided the response into two parts.

A.P4., L123. Please correct for column-averaged abundances of $CO_2$ expression.

The abundance of $CO_2$ is a measure of the occurrence of the chemical elements relative to all other elements in a given environment. Abundance is measured in one of three ways: by the mass-fraction (the same as weight fraction); by the mole-fraction (fraction of atoms by numerical count, or sometimes fraction of molecules in gases); or by the volume-fraction. Volume-fraction is a common abundance measure in mixed gases such as planetary atmospheres, and is similar in value to molecular mole-fraction for gas mixtures at relatively low densities and pressures, and ideal gas mixtures. And the L123 in the previous manuscript was used to introduce information about the TCCON site data, and this expression is cited on the TCCON website (https://tccondata.org/, last access: 10-Nov-2022.). Therefore, the L123 expression is correct and there is no need to correct the expression of L123. And The expression from the TCCON website is shown below.

TCCON is a network of ground-based Fourier Transform Spectrometers recording direct solar spectra in the near infrared spectral region. From these spectra, accurate and precise column-averaged abundances of $CO_2$ are retrieved and reported here.

B.And the results showed that R2 was 0.9686, and RMSE was 1.3811. Please indicate the source.

"And the results showed that $R^2$ was 0.9686, and RMSE was 1.3811." is the comparison result by comparing the predicted data with TCCON for all sites, and this sentence should be located in Section 3.1. Therefore, "And the results showed that $R^2$ was 0.9686, and RMSE was 1.3811 ppm." was removed in the latest manuscript.

11. P5. 146 EBK theory or EBK method? Please express it in a unified way. such as L150.

Dear reviewer, thank you for your kindly suggestions. We have checked this issue in the paper. And The issue has been uniformly revised to EBK method.

12. In Section 2.4, more indicators for accuracy evaluation should be added. Bias and standard deviation are necessary in the verification of $XCO_2$.

Dear reviewer, thank you for your kindly suggestions. The bias ($Bias$) and standard deviation (σ) were added as evaluation indicators in Section 2.4. And the $Bias$ and σ can be calculated as follows:

$$\overline{p} = \frac{1}{N} \sum_{i=1}^{N} P_i, \tag{3}$$

$$Bias = \frac{1}{N} \sum_{i=1}^{N} (P_i - R_i), \tag{5}$$

$$\sigma = \sqrt{\frac{1}{N} \sum_{i=1}^{N} |P_i - \overline{p}|^2}, \tag{6}$$

where $N$ is the number of prediction locations, $P_i$ is the predicted value, and $R_i$ is the observed value.

13. P7. 195. It is not necessary to show the results of each year, only a few examples, such as some months or a specific year, are sufficient. From the analysis of the data, such as some seasonal changes, changes in $CO_2$ growth, and spatial differences may be more meaningful.

Dear reviewer, thank you for your kindly suggestions. The purpose of this paper is to make a dataset, to introduce the contents of the dataset in detail and to verify the accuracy of the dataset. Therefore, this paper does not focus on seasonal analysis and spatial differences. However, considering that seasonal variation analysis can evaluate the reasonableness of the CDC dataset, we produced the distribution of seasonal variation of $XCO_2$ in 2010 in Fig 13-1. Fig. 13-1a shows the mean values of $XCO_2$ data from January to March 2010, Fig. 13-1b from April to June 2010, Fig. 13-1c from July to September 2010, and Fig. 13-1d from October to December 2010.

Besides, the evaluation index of the annual mean growth rate (AMGR) of $XCO_2$ can indirectly verify the accuracy of our product. Therefore, the AMGR of $XCO_2$ were calculated for the different TCCON sites and are presented in the Figure 13-2. Figure 13-2 shows the AMGR of $XCO_2$ from TCCON site and from CDC dataset (The predicted data from this paper) at the 23 TCCON sites. The horizontal axis of the graph represents time and the vertical axis represents the annual mean growth rate (AMGR). The red and blue error bars in the figure represent the standard deviation of the CDC dataset and the TCCON site dataset in the current year data at different sites. For example, the AMGR of $XCO_2$ in 2010 relative to 2009 is plotted at 2009.5 in the figure, and the standard deviation calculated is from the 2010 data. Site names are abbreviated and different subplots are labeled with the abbreviated site names (as shown in Table 1). In particular, the data for all sites were aggregated and the AMGR were calculated, and the AMGR is shown in the subplot labeled 'all'. The results of the site-by-site comparison indicate that the AMGR of CDC is consistent with the AMGR of TCCON, and there is only a slight deviation at a few sites. The reason for this slight deviation may be due to insufficient data. In addition, in the subplot where all data are aggregated (namely, the subplot marked with 'All'), it also shows that the trends of the two indicators (AMGR and standard deviation) are consistent for both CDC data and TCCON data.

[Figure]

**Fig.** 13-1 The distribution of seasonal mean XCO$_2$ in 2010

**Table 1.** Geographic locations of TCCON sites used for validation and the statistics used to compare predicted XCO$_2$ and TCCON XCO$_2$ observations.

| Tccon sites (Site abbreviations) | Longitude | Latitude | $R^2$ | RMSE (ppm) | Mean Bias(ppm) |
|---|---|---|---|---|---|
| Jet Propulsion Laboratory (JC) | -118.18 | 34.20 | 0.98** | 1.07 | -0.89 |
| Caltech (CI) | -118.13 | 34.14 | 0.97** | 0.95 | -1.21 |
| Edwards (DF) | -117.88 | 34.96 | 0.98** | 0.82 | 0.72 |
| Four Corners (FC) | -108.48 | 36.80 | 0.96** | 0.31 | -0.75 |
| Lamont (OC) | -97.49 | 36.60 | 0.98** | 1.04 | -1.79 |
| Park Falls (PA) | -90.27 | 45.94 | 0.98** | 1.24 | -0.62 |
| Manaus (MA) | -60.60 | -3.21 | 0.88** | 0.64 | -1.53 |
| Izana (IZ) | -16.48 | 28.30 | 0.98** | 1.18 | -0.96 |
| Ascension Island (AE) | -14.33 | -7.92 | 0.94** | 0.93 | -0.84 |
| Orléans (OR) | 2.11 | 47.97 | 0.99** | 0.95 | 0.13 |
| Zugspitze (ZS) | 10.98 | 47.42 | 0.92** | 1.52 | -0.40 |
| Garmisch (GM) | 11.06 | 47.48 | 0.98** | 1.05 | 0.36 |
| Nicosia (NI) | 33.38 | 35.14 | 0.93** | 0.73 | -1.38 |
| Réunion Island (RA) | 55.49 | -20.90 | 0.96** | 1.23 | -1.33 |
| Hefei (HF) | 117.17 | 31.90 | 0.87** | 1.51 | -1.82 |
| Burgos (BU) | 120.65 | 18.53 | 0.89** | 1.01 | -1.58 |
| Anmeyondo (AN) | 120.65 | 36.54 | 0.90** | 1.20 | -0.58 |
| Saga (JS) | 130.29 | 33.24 | 0.97** | 1.26 | -1.14 |
| Edwards (DB) | 130.89 | -12.43 | 0.99** | 0.75 | -1.03 |
| Tsukuba (TK) | 140.12 | 36.05 | 0.91** | 1.89 | 0.48 |

| | | | | | |
|---|---|---|---|---|---|
| Rikubetsu (RJ) | 143.77 | 43.46 | 0.95** | 1.17 | 0.19 |
| Wollongong (WG) | 150.88 | -34.41 | 0.99** | 0.82 | -0.58 |
| Lauder01&02&03 (LL) | 169.68 | -45.04 | 0.97** | 1.44 | -0.70 |
| All sites | - | - | 0.97** | 1.38 | -0.65 |

** At the 0.01 level (two-tailed), the correlation is significant.

[Figure]

**Fig.** 13-2 XCO$_2$ growth rate from TCCON site and from CDC dataset (The predicted data from this paper). The horizontal axis of the graph represents time and the vertical axis represents the annual mean growth rate (AMGR).

14.Figure 6, As said at the beginning, of course we know that $CO_2$ is rising, but its growth rate is more meaningful, and it is recommended to draw a related graph of the growth rate. If it does not reflect reasonable fluctuations, but a fully sinusoidal pattern, the study would be significantly flawed.

Dear reviewer, thank you for your kindly suggestions. Therefore, the annual mean growth rate (AMGR) of $XCO_2$ were calculated for the different TCCON sites and are presented in the Figure 14-1. Figure 14-1 shows the $XCO_2$ growth rate from TCCON site and from CDC dataset (The predicted data from this paper) at the 23 TCCON sites. The horizontal axis of the graph represents time and the vertical axis represents the annual mean growth rate (AMGR). The red and blue error bars in the figure represent the standard deviation of the CDC dataset and the TCCON site dataset in the current year data at different sites. For example, the AMGR of $XCO_2$ in 2010 relative to 2009 is plotted at 2009.5 in the figure, and the standard deviation calculated is from the 2010 data. Site names are abbreviated and different subplots are labeled with the abbreviated site names (as shown in Table 1). In particular, the data for all sites were aggregated and the AMGR were calculated, and the AMGR is shown in the subplot labeled 'all'.

Figure 14-1 compares the AMGR of the TCCON and CDC datasets. The results of the site-by-site comparison indicate that the AMGR of CDC is consistent with the AMGR of TCCON, and there is only a slight deviation at a few sites. The reason for this slight deviation may be due to insufficient data. In addition, in the subplot where all data are aggregated (namely, the subplot marked with 'All'), it also shows that the trends of the two indicators (AMGR and standard deviation) are consistent for both CDC data and TCCON data.

**Table 1.** Geographic locations of TCCON sites used for validation and the statistics used to compare predicted $XCO_2$ and TCCON $XCO_2$ observations.

| Tccon sites (Site abbreviations) | Longitude | Latitude | $R^2$ | RMSE (ppm) | Mean Bias(ppm) |
|---|---|---|---|---|---|
| Jet Propulsion Laboratory (JC) | -118.18 | 34.20 | 0.98** | 1.07 | -0.89 |
| Caltech (CI) | -118.13 | 34.14 | 0.97** | 0.95 | -1.21 |
| Edwards (DF) | -117.88 | 34.96 | 0.98** | 0.82 | 0.72 |
| Four Corners (FC) | -108.48 | 36.80 | 0.96** | 0.31 | -0.75 |
| Lamont (OC) | -97.49 | 36.60 | 0.98** | 1.04 | -1.79 |
| Park Falls (PA) | -90.27 | 45.94 | 0.98** | 1.24 | -0.62 |
| Manaus (MA) | -60.60 | -3.21 | 0.88** | 0.64 | -1.53 |
| Izana (IZ) | -16.48 | 28.30 | 0.98** | 1.18 | -0.96 |
| Ascension Island (AE) | -14.33 | -7.92 | 0.94** | 0.93 | -0.84 |
| Orléans (OR) | 2.11 | 47.97 | 0.99** | 0.95 | 0.13 |
| Zugspitze (ZS) | 10.98 | 47.42 | 0.92** | 1.52 | -0.40 |
| Garmisch (GM) | 11.06 | 47.48 | 0.98** | 1.05 | 0.36 |
| Nicosia (NI) | 33.38 | 35.14 | 0.93** | 0.73 | -1.38 |
| Réunion Island (RA) | 55.49 | -20.90 | 0.96** | 1.23 | -1.33 |
| Hefei (HF) | 117.17 | 31.90 | 0.87** | 1.51 | -1.82 |
| Burgos (BU) | 120.65 | 18.53 | 0.89** | 1.01 | -1.58 |
| Anmeyondo (AN) | 120.65 | 36.54 | 0.90** | 1.20 | -0.58 |

| | | | | | |
|---|---|---|---|---|---|
| Saga (JS) | 130.29 | 33.24 | 0.97** | 1.26 | -1.14 |
| Edwards (DB) | 130.89 | -12.43 | 0.99** | 0.75 | -1.03 |
| Tsukuba (TK) | 140.12 | 36.05 | 0.91** | 1.89 | 0.48 |
| Rikubetsu (RJ) | 143.77 | 43.46 | 0.95** | 1.17 | 0.19 |
| Wollongong (WG) | 150.88 | -34.41 | 0.99** | 0.82 | -0.58 |
| Lauder01&02&03 (LL) | 169.68 | -45.04 | 0.97** | 1.44 | -0.70 |
| All sites | - | - | 0.97** | 1.38 | -0.65 |

** At the 0.01 level (two-tailed), the correlation is significant.

[Figure]

**Fig.** 14-1 XCO$_2$ growth rate from TCCON site and from CDC dataset (The predicted data from this paper). The horizontal axis of the graph represents time and the vertical axis represents the annual mean growth rate (AMGR)

15. From the research point of view, averaging kernel and the prior profile should be considered in comparison with OCO-2. Although they may be ignored in some cases and not important on monthly validation where accuracy is not required, the article should mention it.

Dear reviewer, thank you for your kindly suggestions. The averaging kernels indicate the vertical resolution of the measurements and represent the sensitivity of the retrieval to the ''true'' state (Ohyama et al., 2021). But, according to the current research, this difference is about a few tenths of ppmv, which is smaller than their measurement error (Inoue et al., 2013; Liang et al., 2017). Therefore, In the latest manuscript, "The effect of averaging kernel and the prior profile is ignored when comparing results with OCO-2" has been added to Section 3.2.

**References:**

Inoue, M., Morino, I., Uchino, O., Miyamoto, Y., Yoshida, Y., Yokota, T., et al., 2013. Validation of XCO2 derived from SWIR spectra of GOSAT TANSO-FTS with aircraft measurement data. Atmos. Chem. Phys. 13, 9771–9788.

Liang, A.L., Gong, W., Han, G., Xiang, C.Z., 2017. Comparison of satellite-observed XCO2 from GOSAT, OCO-2, and ground-based TCCON. Remote Sens., 9.

Ohyama H, Morino I, Nagahama T, et al. Column-averaged volume mixing ratio of CO2 measured with ground-based Fourier transform spectrometer at Tsukuba[J]. Journal of Geophysical Research: Atmospheres, 2009, 114(D18).

16. Figure 3,4 and 5. PXCO2 TXCO2 P XCO2 should be described uniformly (note the space). Other than that, I would suggest that it would be better to do time series validation on a monthly basis. Specifically, the horizontal axis is time, and the vertical axis is parameters such as error, which can also be filled with error distribution, which is more intuitive.

Dear reviewer, thank you for your kindly suggestions. We have unified the descriptions of PXCO2 TXCO2 and P XCO2 in the full paper. And the P $XCO_2$ and T $XCO_2$ are unified descriptions. In addition, we supplemented the experiments according to the requirements, and the results are shown in Fig. 16-1, Fig. 16-2 and Fig. 16-3.

The Fig. 16-1 shows a group diagram about the monthly-averaged $XCO_2$. The vertical axis of Fig.16-1a is the bias of the calculated monthly-averaged $XCO_2$ through the TCCON values and the CDC dataset at the mid- and low- latitude stations. Fig.16-1b shows the visualized histogram from the data in Fig. 16-1a, where the red line in Fig. 16-1b is the fitted curve of the Gaussian function. The yellow line is the fitted line. Fig. 16-1a show that our bias data are within two standard deviations, and very few points are outside the two standard deviation range. And the mean bias is -0.65pmm for Fig16-1a. The histogram in Fig. b shows that the bias of the CDC data set with the TCCON data is normally distributed with a mean of -0.63 ppm and a standard deviation of 1.4 ppm. Fig. c shows that the $R^2$ of the CDC data set is 0.97 and the RMSE is 1.38 ppm.

Fig.16-2 has the same representation as Fig.16-1, with the difference that Fig.16-2 is comparing the OCO-2 data and the CDC data set. The experimental results showed that the mean bias was -1.57 ppm in Fig. 16-2a. The Gaussian fit model of Fig. 16-2b shows that the mean of the Gaussian model is -1.57 ppm and the standard deviation is 1.40 ppm. Fig.16-2c shows that the

evaluation metric $R^2$ is 0.91 ppm and RMSE is 1.35 ppm for the CDC dataset and the OCO-2 dataset.

Fig.16-3 has the same representation as Fig.16-1, with the difference that Fig.16-3 is a verification of the vacancy strategy. Vacancy strategy means that some data are pre-removed from the original input data. The pre-removed data are used as the validation set to verify the predicted results. The experimental results showed that the mean bias was -0.0043ppm in Fig. 16-3a. The Gaussian fit model of Fig. 16-3b shows that the mean of the Gaussian model is -0.0043ppm and the standard deviation is 0.6842 ppm. Fig.16-3c shows that the evaluation metric $R^2$ is 0.9907 ppm and RMSE is 0.6842 ppm for the CDC dataset and the validation set. **In general, the mean bias of the CDC data set relative to the TCCON, OCO-2, and GOSAT validation sets is -0.65ppm, -1.57ppm, and -0.0043ppm, respectively, with evaluation metrics $R^2$ of 0.97, 0.91, and 0.99.**

[Figure]

Figure 16-1 Monthly-averaged $XCO_2$ validation results for TCCON data and CDC dataset at global mid- and low-latitude TCCON sites from 200906 to 202012.

[Figure]

Figure 16-2 Monthly-averaged $XCO_2$ validation results for OCO-2 data and CDC dataset at global from 2014 to 2020.

[Figure]

Figure 16-3 Monthly-averaged $XCO_2$ validation results based on GOSAT vacancy validation strategy from 200906 to 202012.

17. Figure 7~17. It looks like this resolution may be trapped in a highly smooth phenomenon, which means it may not really be 0.25 degrees. It is recommended to draw a detailed map of some regions to show that the method does have this good resolution and can capture reasonable and sufficient spatial gradient changes. It is also recommended to compare the results of models, such as CarbonTracker or the L4B model products of GOSAT-NIES, to demonstrate the rationality of the results.

Dear reviewer, thank you for your kindly suggestions. Because CarbonTracker is a modeling system, it does not acquire data from direct measurements by satellite or ground station instruments. Besides, we have enough validation experiments from top to bottom (for example, the OCO-2 validation in Section 3.2 and the GOSAT vacancy validation in Section 3.3) and from bottom to top (for example, TCCON data validation in Section 3.1). Moreover, the data for the validation experiments are obtained based on direct measurements from satellites or ground stations in our manuscript. Therefore, there is no need to add new validation experiments in this manuscript. However, we can add a qualitative comparison to demonstrate that the spatial resolution of the CDC dataset is 0.25 degrees and the existence of $XCO_2$ spatial gradient variation in the CDC dataset. Fig. 17-1 shows the distribution of $XCO_2$ in January 2010. The data are from the CarbonTracker (CT2019B version, https://gml.noaa.gov/aftp/products/carbontracker/co2/CT2019B/molefractions/co2_total_monthly/, Last Access:2022/11/15) dataset in Fig. a and b. And Fig. b is a zoom in for a specific region of Fig. a. The data are from the CDC dataset in Fig. c and d. And Fig. d is a zoom in for a specific region of Fig. c. We can learn from Figure d that the spatial resolution of the CDC dataset is 0.25 degrees and there is a spatial gradient variation.

[Figure]

**Figure 17-1** $XCO_2$ distribution maps in January 2010, Fig. a and b show global and local $XCO_2$ maps from CarbonTracker data (CT2019B version), and Fig. c and d show global and local $XCO_2$ maps from the CDC dataset.

18. As mentioned above, in addition to the error evaluation in the time dimension, the error in the spatial dimension should also be evaluated to illustrate the reliability of the data.

Dear reviewer, thank you for your kindly suggestions. CarbonTracker data are simulated data and not observed data based on satellites or ground stations. Therefore, Carbon Tracker data as a validation dataset will have a large uncertainty in the validation results. Although there is a large uncertainty in this validation result, it still has a reference value for evaluating the distribution of spatial errors. The CT2019B version of CarbonTracker data was used and can be downloaded from the official website. (https://gml.noaa.gov/aftp/products/carbontracker/co2/CT2019B/molefractions/co2_total_ monthly/, Last Access:2022/11/18).

There is a difference in data resolution, namely 3°by 2° for the CarbonTracker data and 0.25°by0.25° for the CDC dataset. Thus, the resolution of the CDC dataset was resampled from 0.25°by0.25° to 3°by 2°. And multiple pixel values are averaged is the method of CDC data resampling. Besides, it is worth noting that CarbonTracker data is 25 layers at different heights. Therefore, the CarbonTracker data were averaged directly based on the different height layers data. Figure 18-1 shows the calculated bias based on the CarbonTracker data and the CDC dataset in January 2010. From Figure 18-1, we can see that the bias is [-2.5,2.5] in most of the area, and the bias is more than 2.5 ppm in a few areas. Furthermore, the mean bias is 0.1202 ppm. Therefore, the spatial bias distribution of the CDC data is within a reasonable range.

[Figure]

**Figure 18-1** Spatial bias between CDC dataset and CarbonTracker data (CT2019B version) in January 2010.

19. I am not a Native English speaker. But I believe that the English of this article should be greatly improved, including scientific names of many nouns, and descriptions in scientific language.

Buchwitz, Michael, et al. "Computation and analysis of atmospheric carbon dioxide annual mean growth rates from satellite observations during 2003–2016." Atmospheric Chemistry and Physics 18.23 (2018): 17355-17370.

Chatterjee, A., et al. "Influence of El Niño on atmospheric CO2 over the tropical Pacific Ocean: Findings from NASA's OCO-2 mission." Science 358.6360 (2017): eaam5776.

Dear reviewer, thank you for your kindly suggestions. We checked and revised the language, professional nomenclature, and units in charts and graphs in the paper. In addition, we have read the literature mentioned in your question.